# ONLINE FITTING CONNECTOME-CONSTRAINED DROSOPHILA WHOLE-BRAIN MODEL REPRODUCES CRITICAL RESTING-STATE DYNAMICS

## ABSTRACT

The rapid growth of large-scale synaptic connectome maps and neural activity datasets has created an urgent need for connectome-constrained whole-brain models that can fit and interpret experimentally recorded neural data. A promising approach to bridge this gap is to train biologically inspired models using backpropagation through time (BPTT), which enables data-driven optimization of unknown model parameters. However, BPTT is inherently an offline method, with memory requirements that grow linearly with simulation time, making it impractical for training large-scale whole-brain networks over biologically relevant timescales. To address this challenge, we introduce an online learning framework for fitting whole-brain models using online gradient-based optimization. By updating parameters in a strictly forward-time manner, our method reduces memory consumption to a single time step, scaling only with the number of parameters rather than the entire temporal sequence. Using this framework, we construct a *Drosophila* whole-brain network comprising over 130,000 neurons and millions of synapses, where the network topology is fixed from the FlyWire connectome, and unknown parameters such as synaptic weights and cellular time constants are optimized to match *in vivo* resting-state neural activity. Our results show that this approach enables the training of large-scale *Drosophila* models over experimental timescales on a single GPU, a feat that is computationally prohibitive with BPTT. Remarkably, the optimization not only captures target dynamics but also spontaneously produces synaptic weight distributions that closely match empirical connectome statistics and drives the network toward a hallmark feature of resting state – critical dynamics. Together, this work establishes an online, scalable, and data-driven framework for integrating anatomical and functional datasets, paving the way toward mechanistic whole-brain models at unprecedented scales.

## 1 INTRODUCTION

The quest to understand how the brain's structural connectivity gives rise to its complex dynamics represents one of the fundamental challenges in neuroscience. Recent technological advances have produced unprecedented datasets documenting both the anatomical wiring and functional activity of neural systems (Amunts et al., 2013; Cook et al., 2019; Dorkenwald et al., 2024; Oh et al., 2014; Ding et al., 2025). The complete synaptic-resolution connectome of *Drosophila*, comprising over 130,000 neurons and millions of synapses, now provides a comprehensive structural blueprint of an entire fly brain (Dorkenwald et al., 2024). In parallel, large-scale calcium imaging and electro-physiological recordings have captured the rich spatiotemporal dynamics of neural activity across whole-brain populations (Mann et al., 2017; Turner et al., 2021). However, a critical gap remains: we lack computational frameworks capable of integrating these anatomical and functional datasets into mechanistic models that can explain how connectivity patterns shape neural dynamics.

Current approaches to modeling neural activity fall into two distinct paradigms, each with fundamental limitations. Task-optimized artificial neural networks (ANNs) have revolutionized our ability to predict neural responses across diverse brain systems. Deep convolutional networks trained for object recognition in macaque inferotemporal cortex, with improved task performance correlating with higher neural predictivity (Yamins et al., 2014; Nayebi et al., 2018). Despite their impressive

predictive accuracy, these models suffer from critical shortcomings in biological interpretability. Without incorporating known anatomical constraints or biological mechanisms, they typically operate as black-box function approximators that can predict what neurons will do but not why specific representations emerge or how they relate to known circuit mechanisms.

At the opposite extreme, detailed biophysical models prioritize biological realism by incorporating precise anatomical connectivity and physiological properties. Recent efforts have successfully integrated multi-modal experimental data to construct models with tens of thousands of neurons, reproducing visual responses in the mouse cortex (Billeh et al., 2020; Chen et al., 2022) and motion detection in the *Drosophila* visual system (Lappalainen et al., 2024). These models respect the constraints imposed by synaptic-resolution connectomes and train unknown parameters such as synaptic weights and physiological parameters using the backpropagation through time (BPTT) algorithm. However, this detailed modeling paradigm faces insurmountable computational barriers when scaling to whole-brain systems. BPTT requires storing complete activation histories throughout training, with memory consumption that scales linearly with both network size and simulation duration. Training a 50,000-neuron mouse visual cortex model required 160 GPUs, and even then only for relatively short behavioral tasks (Chen et al., 2022). For whole-brain networks comprising hundreds of thousands of neurons operating over biologically relevant timescales, BPTT becomes computationally prohibitive. This fundamental limitation has created an impasse: researchers must choose between biologically grounded models that cannot scale (Zhu et al., 2025) and scalable models that lack biological grounding (Pathak et al., 2022).

In this work, we introduce an online learning framework to resolve this computational bottleneck. Rather than accumulating gradients across entire trajectories, our method updates parameters in a strictly forward-time manner, reducing memory requirements from scaling with simulation length to depending only on the number of model parameters. This approach makes it feasible, for the first time, to optimize large-scale brain networks over experimental timescales on standard computational resources. We demonstrate this framework by constructing a whole-brain *Drosophila* model where the complete FlyWire connectome provides the fixed anatomical scaffold (Dorkenwald et al., 2024), while unknown parameters, including synaptic weights and cellular time constants, are optimized to match experimentally recorded neural dynamics (Mann et al., 2017; Turner et al., 2021).

Our results reveal that this data-driven optimization process not only successfully reproduces target neural activity patterns but also yields emergent properties that were not explicitly enforced. The optimized synaptic weight distributions closely match those observed in the empirical connectome, suggesting that functional constraints shape synaptic strengths in predictable ways. Moreover, the trained network spontaneously develops complex dynamical features characteristic of biological neural systems. These findings demonstrate that online gradient-based optimization can serve as a powerful bridge between structure and function, enabling whole-brain models that are simultaneously constrained by anatomy, fitted to experimental data, and capable of revealing principles of neural organization.

## 2 RELATED WORK

**Neural activity fitting.** Modeling neural activity through network fitting has become an important tool in neuroscience. One widely used approach represents a brain region as a probabilistic recurrent spiking network, with parameters optimized to maximize the likelihood of observed spike trains (Gerwinn et al., 2010; Gerhard et al., 2013). Although effective for capturing certain statistical features of neural responses, these models often fail to reproduce realistic large-scale population dynamics and circuit-level activity patterns (Bellec et al., 2021). Since 2014, deep learning has catalyzed a complementary line of work that uses task-optimized ANNs as brain models. Notably, Yamins, DiCarlo, and colleagues showed that deep convolutional networks trained for object recognition yield intermediate representations that closely track macaque IT responses (Yamins et al., 2014; Nayebi et al., 2018), with better task performance accompanying higher neural predictivity (Kubilius et al., 2019; Schrimpf et al., 2020). This task-driven paradigm has broadened beyond the visual system: sequence-to-sequence RNNs have been used to model sentence-level responses in language areas (Hosseini et al., 2024); speech recognition networks help predict auditory-cortex encoding (Ahmed et al., 2025); and deep reinforcement-learning agents sometimes develop grid-like codes reminiscent of hippocampal representations (Banino et al., 2018). However, ANN models face

challenges in biological interpretability. They often act as black boxes, accurately predicting neural responses without explaining why certain representations emerge or how they relate to real circuits. Lacking anatomical, physiological, and biophysical constraints, their correspondence to biological neurons is ambiguous, and apparent similarities to neural data may reflect task-driven coincidences rather than true mechanisms.

**Connectome-constrained brain modeling.** The field of connectomics has progressively mapped neural wiring across species, from the complete nervous system of *C. elegans* (Cook et al., 2019) to the synaptic-resolution connectome of *Drosophila* (Dorkenwald et al., 2024) and emerging maps of mouse (Oh et al., 2014; Ding et al., 2025) and human brains (Amunts et al., 2013). These resources have opened the door to connectome-constrained modeling, where the anatomical scaffold of the connectome is used to build computational models that link structure to function. At the macroscopic scale, whole-brain models using neural mass models prioritize scale over detail by abstracting each brain region as a neural mass (Pathak et al., 2022). These models use anatomical connectivity from diffusion MRI as structural scaffolding, with nodes representing mean-field activity of brain areas that evolve according to differential equations under coupling influences from other regions (Griffiths et al., 2021). While computationally tractable at whole-brain scales, this approach suffers from severely limited predictive resolution. More critically, these models typically rely on only a few global coupling parameters, fundamentally constraining their capacity for individualized prediction and task generalization. At the cellular scale, neuron-resolved models pursue the opposite strategy, prioritizing biological detail at the expense of scale. Recent advances have demonstrated remarkable success in this direction: Billeh et al. (2020) systematically integrated multimodal data to create biologically realistic models of mouse primary visual cortex with over 50,000 neurons, which Chen et al. (2022) successfully trained on visual tasks using backpropagation across 160 GPUs. Similarly, Lappalainen et al. (2024) trained a *Drosophila* visual system model constrained by synapse-level connectome data, reproducing motion detection and experimental neural responses without requiring precise physiological parameters for individual neurons. Zhu et al. (2025) further demonstrated how optimized barrel cortex models can replicate network dynamics during whisker processing. However, this detailed modeling paradigm faces critical limitations that prevent its extension to whole-brain scales. Scaling BPTT for whole-brain-scale modeling is exceedingly difficult due to its massive computational resource demands(Lillicrap et al., 2020). More practically, the memory and computational costs grow prohibitively with simulation length, rendering activity fitting over biological time scales infeasible for large-scale networks.

**Critical state of resting-state neural activity.** Extensive experimental studies across multiple species consistently demonstrate that resting-state neural activity in the brain exhibits hallmark characteristics of critical systems, such as neuronal avalanches with power-law distributions (Beggs & Plenz, 2003; Ponce-Alvarez et al., 2018; Fuscà et al., 2023; Fontenele et al., 2019). These observations raise two fundamental questions: Why does the resting brain tend to operate near criticality? And how does the neural system spontaneously maintain such delicate dynamical balance? A prevailing view is that criticality is the outcome of functional optimization (O'Byrne & Jerbi, 2022). Operating at the critical point provides the unique computational advantages: it enables the brain to elegantly navigate between disordered random states and overly synchronized states, establishing an optimal balance between stability and flexibility while simultaneously maximizing information transmission capacity and processing efficiency (Shew & Plenz, 2013; Tkačik et al., 2015). To test this hypothesis, researchers have developed diverse computational models to reproduce critical-state features of neural activity. At the network level, studies using excitatory-inhibitory neural networks with random graphs have shown that when network parameters (such as connection strength and excitatory-inhibitory balance) are precisely tuned near a phase transition point, the system can faithfully reproduce experimentally observed power-law distributed neuronal avalanches and long-range correlated dynamics (Shew et al., 2009). At the statistical level, Ising models abstract the brain as a spin network with phase transition behavior. Near the critical temperature, these models exhibit multi-scale fluctuation patterns and complex correlation structures that closely resemble those of the biological brain (Nicoletti et al., 2020; Cabral-Carvalho et al., 2025). Despite these advances, existing models are overly simplified, focusing on statistical patterns like power-law distributions while neglecting the prediction of complete neural signal dynamics. They also rely on idealized network topologies rather than real brain anatomical connectivity and require manual parameter tuning to maintain criticality. These shortcomings highlight the need for biologically realistic models that can self-organize to sustain criticality.

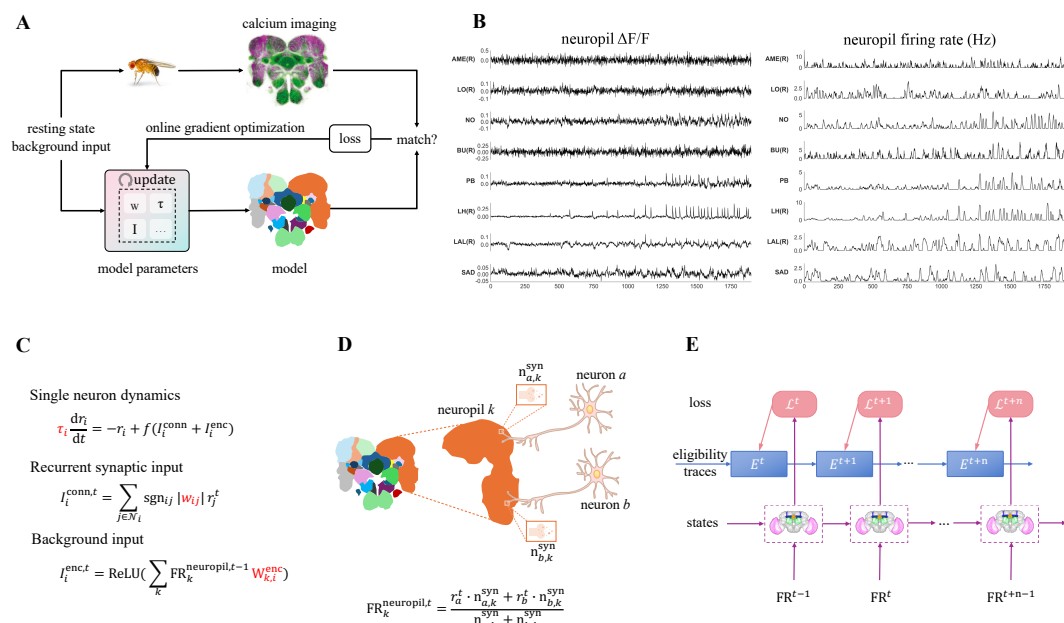

Figure 1: **Online fitting workflow of *Drosophila* whole-brain neural activity.** (A) Online fitting paradigm, which employs online gradient optimization to minimize the discrepancy between network activity and experimental data. (B) Example traces of neuropil calcium signals $\Delta F/F$ and corresponding firing rate conversions. (C) Single-neuron dynamics with free parameters (highlighted in red) optimized during training. (D) Neuropil firing-rate readout constrained by the synapse-resolved connectome. (E) Online optimization diagram during whole-brain neural activity fitting.

## 3 ONLINE FITTING OF *Drosophila* WHOLE-BRAIN NEURAL ACTIVITY

### 3.1 ONLINE FITTING FRAMEWORK

We develop an online fitting framework that bridges anatomical structure and functional dynamics by integrating connectome constraints with scalable gradient-based algorithms to model whole-brain *Drosophila* neural activity (Fig. 1A). The workflow begins with large-scale calcium imaging data acquired from *Drosophila* brains during spontaneous resting states (Mann et al., 2017; Turner et al., 2021), which provide ground-truth spatiotemporal patterns of neural activity that our biological model must reproduce. To build the model, we use the FlyWire synaptic-resolution connectome (Dorkenwald et al., 2024) as the fixed anatomical scaffold and describe the dynamics of each neuron with firing-rate threshold-linear models (Miller & Fumarola, 2012). While the connectome specifies the network topology, several biophysical parameters remain unknown, including synaptic weights, neuronal time constants, and background input strengths. These parameters are treated as free variables to be optimized against experimental activity data. Model optimization is performed with an online gradient optimization algorithm implemented in the BrainScale platform (Wang et al., 2024). Unlike BPTT, our online approach scales in memory only with the number of model parameters, making it computationally feasible to fit a whole-brain, connectome-constrained model directly to experimental recordings over biologically relevant timescales.

### 3.2 WHOLE-BRAIN CALCIUM IMAGING NEURAL ACTIVITY

Whole-brain *in vivo* calcium imaging datasets of resting-state activity in *Drosophila* have been collected across 18 brains (Mann et al., 2017; Turner et al., 2021). Recordings were acquired at a sampling rate of 1.2 Hz and subsequently registered to a standardized template brain atlas comprising 73 neuropils (Jenett et al., 2012). The resulting data are represented as $\Delta F/F$ fluorescence signals for each neuropil, providing population-level activity traces across the entire brain (Fig. 1B).

To enable direct comparison with model outputs, which are expressed in firing rates (section 3.3), the calcium fluorescence signals were transformed into estimated firing-rate dynamics. This conversion was performed using a sparse deconvolution method (Appendix A), which infers underlying neuronal activity from the slower calcium signal dynamics. The resulting dataset thus provides neuropil-level firing-rate activity patterns that serve as training targets for our whole-brain connectome-constrained model.

### 3.3 SINGLE-NEURON DYNAMICS

The activity of each neuron $i$ is modeled by a first-order rate equation (Fig. 1C):

$$\tau_i \frac{\mathrm{d}r_i}{\mathrm{d}t} \; = \; -r_i \; + \; f(I_i^{\mathrm{conn}} + I_i^{\mathrm{enc}}) \,, \tag{1}$$

where $r_i$ denotes the firing rate of neuron $i$, and $\tau_i$ is its membrane time constant. The term $I_i^{\mathrm{conn}}$ represents the recurrent synaptic input arising from the connectome-defined network, while $I_i^{\mathrm{enc}}$ denotes external driving input, such as background or stimulus-related signals. The nonlinear activation function $f(\cdot)$ determines how inputs are transformed into firing-rate responses. To ensure firing rates remain non-negative while avoiding artificial saturation effects, we employ the rectified linear function: $f(x) = \max(0, x)$. This choice provides a simple yet effective approximation of neuronal response functions, consistent with threshold-linear models widely used in theoretical neuroscience (Miller & Fumarola, 2012).

To simulate the model, we use the exponential Euler approximation with time step $\Delta t$:

$$r_i(t) = \alpha_i r_i(t - 1) + (1 - \alpha_i) f(I_i^{\mathrm{conn}}(t) + I_i^{\mathrm{enc}}(t)) \,, \tag{2}$$

where the decay factor is given by $\alpha_i = \exp(-\Delta t / \tau_i)$ and is learned during optimization.

### 3.4 CONNECTOME-BASED RECURRENT CONNECTION

Neurons are interconnected according to the synaptic-resolution *Drosophila* connectome provided by the FlyWire project (Dorkenwald et al., 2024; Schlegel et al., 2024). We used the version 783 release, which contains 138,639 neurons and 15,091,982 synaptic connections. In this work, we incorporated two key features from the connectome: (i) the binary connectivity structure, defined as the set of presynaptic partners $\mathcal{N}_i$ for each neuron $i$, and (ii) the synaptic polarity of each connection, $\mathrm{sgn}_{ij}$. Polarity was assigned based on neurotransmitter identity: acetylcholine and dopamine were considered excitatory (+1), while GABA, glutamate, octopamine, and serotonin were considered inhibitory (−1), consistent with their dominant physiological effects in the fly brain. Therefore, the recurrent synaptic input to neuron $i$ at time $t$ is thus expressed as (Fig. 1C):

$$I_i^{\mathrm{conn}}(t) \; = \; \sum_{j \in \mathcal{N}_i} \mathrm{sgn}_{ij} \, |w_{ij}| \, r_j(t),$$

where $r_j(t)$ is the firing rate of presynaptic neuron $j$, $\mathrm{sgn}_{ij} \in \{+1, -1\}$ denotes synaptic polarity, and $|w_{ij}|$ represents the magnitude of the effective synaptic weight. Synaptic weights are initialized randomly (Appendix B) and subsequently optimized during model fitting.

### 3.5 RESTING-STATE BACKGROUND INPUT

While our model explicitly captures direct synaptic connections through the connectome, numerous sources of input remain unaccounted for during resting states. To capture these unknown input influences without explicitly modeling each mechanism, we introduce a data-driven background input term that allows the network to generate realistic spontaneous activity patterns. Specifically, we model the background input $I_i^{\mathrm{enc}}(t)$ to each neuron as a learned function of the population-level activity state (Fig. 1C):

$$I_i^{\mathrm{enc}}(t) \; = \; \mathrm{ReLU}\left( \sum_k \mathrm{FR}_k^{\mathrm{neuropil}}(t - 1) \, W_{k,i}^{\mathrm{enc}} \right), \tag{3}$$

where $\mathrm{FR}_k^{\mathrm{neuropil}}(t - 1)$ denotes the firing rate of neuropil $k$ at time $t - 1$, and $W_{k,i}^{\mathrm{enc}}$ represents the effective coupling strength from neuropil $k$ to neuron $i$. The ReLU nonlinearity ensures non-negative input values, consistent with excitatory drive. This formulation reflects the hypothesis that

each neuron's background input is shaped by large-scale population activity at the neuropil level, thereby coupling single-neuron dynamics to global brain-wide activity patterns.

### 3.6 NEUROPIL FIRING RATE READOUT

Our model operates at single-cell resolution with over 130,000 individual neurons, while the calcium imaging data are spatially averaged within anatomically defined neuropil regions. To enable quantitative comparison between model predictions and experimental observations, we need to transform single-neuron firing rates into neuropil-level population activity. We leverage the anatomical organization of the *Drosophila* brain, where neurons extend processes across multiple neuropils, forming region-specific synaptic territories. Specifically, we compute the population firing rate for each neuropil by aggregating contributions from all neurons that form synapses within that region, weighted by their relative synaptic density (Fig. 1D). Formally, the firing rate $\mathrm{FR}_k^{\mathrm{neuropil}}(t)$ of neuropil $k$ at time $t$ is computed as:

$$\mathrm{FR}_k^{\mathrm{neuropil}}(t) = \frac{\sum_{j \in \mathcal{M}_k} r_j(t) \cdot n_{j,k}^{\mathrm{syn}}}{\sum_{j \in \mathcal{M}_k} n_{j,k}^{\mathrm{syn}}}, \tag{4}$$

where $\mathcal{M}_k$ denotes the set of all neurons with presynaptic terminals in neuropil $k$, and $n_{j,k}^{\mathrm{syn}}$ quantifies the number of synaptic connections that neuron $j$ forms within neuropil $k$. This synapse-weighted averaging reflects the principle that neurons with more extensive arborizations in a given neuropil contribute more strongly to the calcium signal measured from that region.

### 3.7 NEURAL ACTIVITY FITTING WITH ONLINE GRADIENT OPTIMIZATION

We train the *Drosophila* whole-brain model using the D-RTRL online algorithm implemented in BrainScale (Wang et al., 2024). Unlike BPTT, D-RTRL computes gradients in a strictly forward-time manner by propagating eligibility traces that accumulate local parameter sensitivities at each time step. This makes the algorithm both scalable and memory-efficient.

For recurrent weights $w_{ij}$, the gradient is given by:

$$\nabla_{w_{ij}} \mathcal{L} \approx \sum_t \partial \mathcal{L}(t)/\partial r_i(t) \cdot \epsilon_{w_{ij}}(t), \tag{5}$$

$$\epsilon_{w_{ij}}(t) = \alpha_i \cdot \epsilon_{w_{ij}}(t-1) + (1-\alpha_i) \cdot f'(x_i(t)) \cdot \mathrm{sgn}_{ij} \cdot r_j(t). \tag{6}$$

For encoding weights $W_{k,i}^{\mathrm{enc}}$, the gradient is computed by:

$$\nabla_{W_{k,i}^{\mathrm{enc}}} \mathcal{L} \approx \sum_t \partial \mathcal{L}(t)/\partial r_i(t) \cdot \epsilon_{W_{k,i}^{\mathrm{enc}}}(t), \tag{7}$$

$$\epsilon_{W_{k,i}^{\mathrm{enc}}}(t) = \alpha_i \cdot \epsilon_{W_{k,i}^{\mathrm{enc}}}(t-1) + (1-\alpha_i) \cdot f'(x_i(t)) \cdot g'(y_i(t)) \cdot \mathrm{FR}_k^{\mathrm{neuropil}}(t-1). \tag{8}$$

For decay factors $\alpha_i$, which control temporal integration, the gradient is given by:

$$\nabla_{\alpha_i} \mathcal{L} \approx \sum_t \partial \mathcal{L}(t)/\partial r_i(t) \cdot \epsilon_{\alpha_i}(t), \tag{9}$$

$$\epsilon_{\alpha_i}(t) = \alpha_i \cdot \epsilon_{\alpha_i}(t-1) + r_i(t-1) - f(x_i(t)). \tag{10}$$

Here, $x_i(t) = I_i^{\mathrm{conn}}(t) + I_i^{\mathrm{enc}}(t)$ is the total input to neuron $i$, $y_i(t) = \sum_k \mathrm{FR}_k^{\mathrm{neuropil}}(t-1) \cdot W_{k,i}^{\mathrm{enc}}$ is the pre-ReLU encoding input, $f'(\cdot)$ is the derivative of the activation function $f$, $g'(y) = \mathbf{1}_{y>0}$ is the derivative of the ReLU function, $\epsilon_\theta(t)$ represents the eligibility trace for parameter $\theta$, and the loss $\mathcal{L}$ is defined as the mean squared error between recorded neural activity and simulated activity.

During training, the network states and all eligibility traces evolve in real time, updated locally at each step without storing the full temporal history (Fig. 1E and Fig. S8). Once the loss is computed, parameter gradients are obtained directly from the current eligibility traces.

## 4 ONLINE FITTED WHOLE-BRAIN MODELS RECOVER RESTING-STATE CRITICALITY

### 4.1 CONNECTOME-CONSTRAINED ONLINE OPTIMIZATION ENABLES WHOLE-BRAIN FITTING

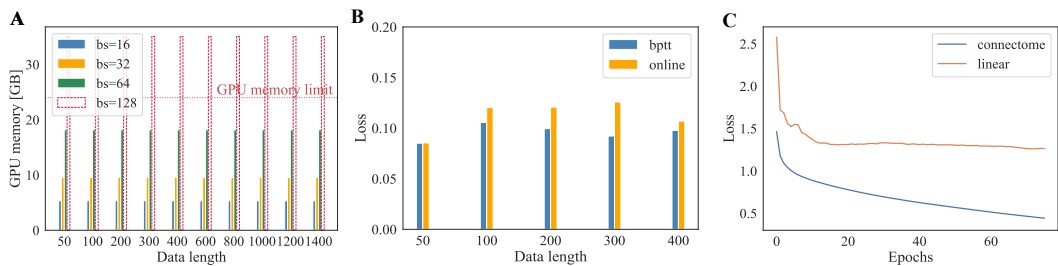

Figure 2: **Scalability and training performance of the online fitting workflow.** (A) GPU memory consumption of our online fitting algorithm across different data lengths and batch sizes. (B) Training loss comparison between the online fitting algorithm and BPTT when optimizing low-rank recurrent weights. (C) Training loss comparison between connectome-constrained readout and linear readout across training epochs.

We first assessed the scalability of our *Drosophila* online fitting framework. We found that even with a small training batch, the memory consumption of BPTT exceeded the capacity of a 32 GB GPU, and training failed regardless of the dataset length. In contrast, our online learning approach successfully trained the model while exhibiting favorable scaling properties. Specifically, memory usage grew linearly with the training batch size but remained independent of the dataset length (Fig. 2A). This property highlights a key advantage of our framework: by decoupling memory demand from sequence length, it enables fitting of large-scale whole-brain networks over biologically realistic timescales.

We next evaluated the training performance of our online fitting approach in comparison with BPTT. To ensure a fair comparison, we replaced the connectome-derived recurrent weight matrix with a synthetic low-rank factorization (Appendix C). Under this setting, BPTT was able to successfully train on a single GPU. We trained models for 200 epochs across datasets of varying lengths and compared their final training losses. The results show that our online learning approach achieves training performance comparable to BPTT (Fig. 2B), while retaining its memory-efficiency advantages (SI).

Having established the computational efficiency of our online fitting framework, we next examined the role of biological constraints in shaping model performance. Specifically, we compared our connectome-based readout mechanism (section 3.6) with an unconstrained linear readout (Appendix D). We found that the connectome-constrained readout consistently converges to a lower training loss, whereas the purely linear readout fails to do so (Fig. 2C). This result highlights the critical role of anatomical priors, which provide an inductive bias that enables the model to more accurately capture the brain's functional dynamics.

### 4.2 TRAINED MODEL REPRODUCES NEURAL DYNAMICS AND FUNCTIONAL CONNECTIVITY

We trained the connectome-constrained model to reproduce neural activity over the first 500 time steps ("Train" phase in Fig. 3A). To assess its capacity for generalization beyond the training window, we simulated the model over longer timescales and compared its predicted activity with unseen experimental data. Remarkably, the trained model was able to generate spontaneous oscillatory dynamics that closely matched those observed during training ("Test" phase in Fig. 3A). The oscillation patterns were preserved across neuropil populations, and the model reproduced characteristic fluctuations in activity, including rising and falling phases evident in the experimental recordings (see markers **a** and **b** in Fig. 3A). These results demonstrate that the model captures intrinsic resting-state dynamics rather than overfitting to the training segment.

Beyond reproducing individual neuropil activity, we examined whether the model also preserved coordinated interactions among neuropils across the whole brain. To this end, we evaluated functional connectivity (FC) patterns during both training and testing phases. On the training data, the

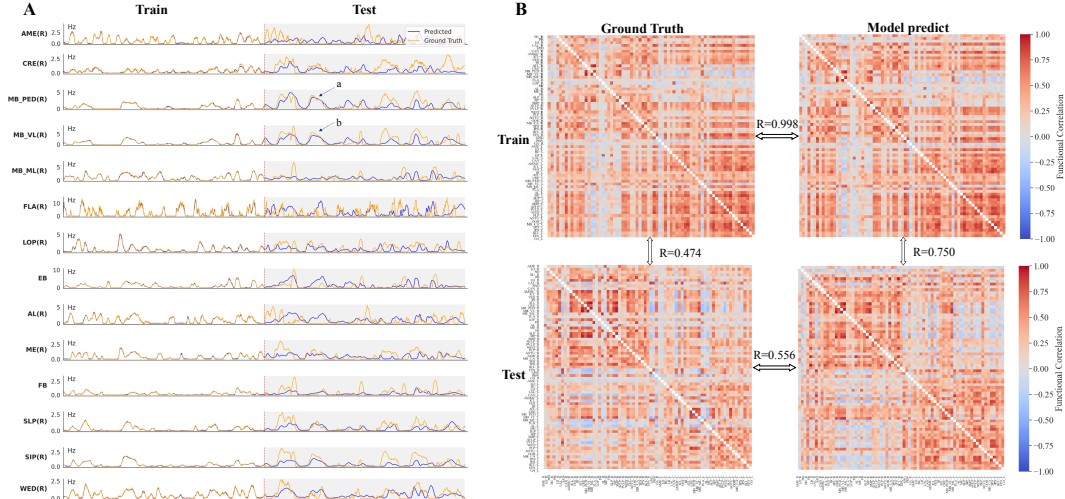

Figure 3: **Neural activity and functional connectivity in trained *Drosophila* whole-brain models.** (A) Simulated neural activity generated in the whole-brain model trained on the first 500 time steps and tested on the subsequent 500. (B) Comparison between ground-truth and model-predicted functional activity across 73 neuropils.

model achieved near-perfect reconstruction of FC, with a correlation of 0.998 relative to the ground-truth experimental connectivity (Fig. 3B). Crucially, during testing on unseen data, the predicted FC remained strongly aligned with empirical measurements (correlation = 0.556), outperforming the direct similarity between the experimental test and training datasets themselves (correlation = 0.474). Furthermore, the model's own predicted activity exhibited robust temporal consistency, as reflected by a correlation of 0.750 between training and test segments of the generated sequences.

## 4.3 TRAINED SYNAPTIC WEIGHTS ALIGN WITH CONNECTOME STATISTICS

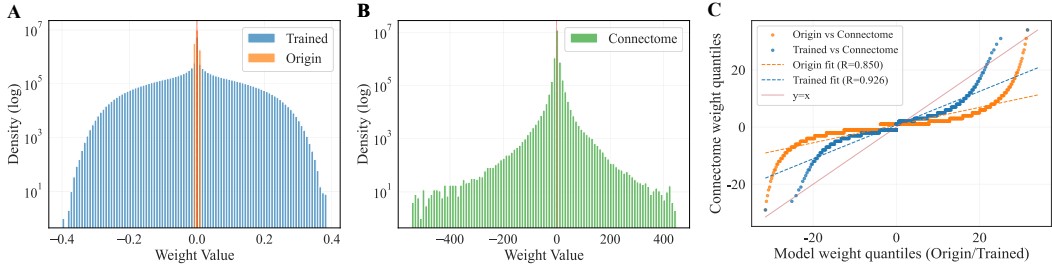

Figure 4: **Recurrent Weight distribution of trained whole-brain models.** (A) Distribution of recurrent weights in models before and after training. (B) Distribution of recurrent weights in the FlyWire *Drosophila* connectome. (C) Q-Q plot comparisons of weight distributions between untrained and trained models relative to the connectome.

Next, we analyzed the distribution of recurrent synaptic weights in the trained whole-brain *Drosophila* models. Before training, weights were narrowly distributed within a limited range (Fig. 4A). After training, the distribution broadened and developed a pronounced heavy tail, with a small subset of connections becoming substantially stronger (Fig. 4A). This organization, with many weak connections and few strong ones, mirrors the empirical pattern observed in the FlyWire connectome reconstruction (Dorkenwald et al., 2024), where synaptic connection counts span several orders of magnitude and a minority of connections dominate in strength (Fig. 4B).

To quantitatively assess similarity, we rescaled both pre- and post-training model weights to the same range as the experimental connectome and performed quantile–quantile (Q–Q) analysis. The post-training weights exhibited a markedly higher correlation with the connectome distribution compared

to the initial weights (Fig. 4C). In the Q–Q plot, points for the trained model clustered closely along the $y = x$ reference line, indicating that both the variability and quantile structure of the learned distribution strongly matched the experimental data. This alignment was further corroborated by higher correlations in the binned count distributions of weight values (Fig. S9).

### 4.4 NETWORK DYNAMICS EXHIBIT CRITICALITY IN THE RESTING STATE

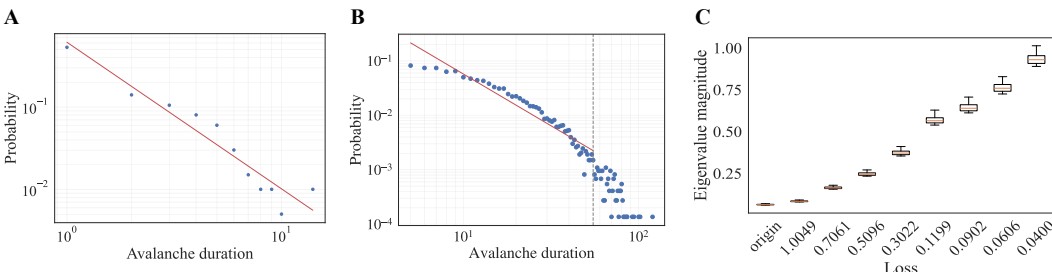

Figure 5: **Analysis of critical dynamics in experimental data and model predictions.** (A) Power-law distribution of avalanche durations in experimentally recorded neural activity, with fitted exponent $\alpha = 1.78$ ($R^2 = 0.916$). (B) Power-law distribution of avalanche durations in model-predicted activity from 138,639 neurons, with fitted exponent $\alpha = 1.90$ ($R^2 = 0.912$). (C) Evolution of the magnitudes of the top 1000 eigenvalues as the training decreases.

A hallmark of resting-state neural activity is criticality, characterized by neuronal avalanches whose duration distribution follows a power law, $P(D) \propto D^{-\alpha}$. Across species, critical states are associated with an exponent $\alpha$ close to 2, as reported in zebrafish (Ponce-Alvarez et al., 2018), rat cortex (Beggs & Plenz, 2003), awake macaque (Gireesh & Plenz, 2008), and humans (Shriki et al., 2013). Using the experimental calcium imaging data (Mann et al., 2017; Turner et al., 2021), we confirmed that the resting-state *Drosophila* brain also operates near criticality, exhibiting a power-law distribution of avalanche durations with a fitted exponent of $\alpha = 1.78$ ($R^2 = 0.916$; Fig. 5A).

Recording avalanche dynamics at single-neuron resolution across the entire brain is technically challenging in experiments. Our model provides a way to overcome this limitation. We analyzed the firing rates of 138,639 neurons simulated over 1,400 time steps from a well-trained model (final loss $\approx 0.03$). Importantly, the last 900 steps, unseen during training, were used for analysis. The model robustly reproduced avalanche criticality, with a fitted exponent of $\alpha = 1.90$ ($R^2 = 0.912$; Fig. 5B).

To further probe the mechanistic basis of this phenomenon, we examined the spectral radius of the recurrent connectivity matrix during training. As optimization progressed and the training loss decreased, the spectral radius increased and asymptotically approached 1 (Fig. 5C). In dynamical systems theory, a spectral radius near unity signifies operation at the edge of stability and near criticality. These results demonstrated that online optimization not only improved data fitting but also spontaneously drove the network from a stable initial regime toward critical dynamics, enabling the model to replicate a fundamental feature of resting-state brain activity.

## 5 CONCLUSION

In conclusion, we presented a connectome-constrained, online learning framework that makes whole-brain data fitting computationally tractable at cellular resolution. By updating parameters strictly in forward time, our method reduces memory from scaling with sequence length to scaling only with the number of parameters, enabling optimization of a *Drosophila* whole-brain network with over 130,000 neurons and millions of synapses on a single GPU. Trained on resting-state calcium recordings, the model reproduces neuropil-level dynamics and functional connectivity on held-out data. Moreover, optimization yielded emergent structure–function alignment: Post-training synaptic weights developed heavy-tailed statistics that more closely match the connectome measured spine counts, and the network's dynamics self-organized toward a critical regime. These results indicate that fitting to whole-brain activity, under anatomical constraints, can recover organizing principles of neural computation, and link circuit topology, parameter distributions, and population dynamics within a single mechanistic model.

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

## A    CONVERTING CALCIUM IMAGING DATA INTO FIRING RATE

The observed calcium signal $c(t)$ is modeled as the convolution of an unobserved spike train $s(t)$ (where $s(t) \in 0, 1$) with an exponential calcium response kernel $k(t) = A \cdot exp(-t/\tau)$ for $t \geq 0$, plus additive noise $\varepsilon(t)$:

$$c(t) = \sum_{\tau} s(\tau) \cdot k(t - \tau) + \varepsilon(t). \tag{S1}$$

To recover the firing rate, we first solved for the most likely spike sequence $\hat{s}(t)$ given the calcium trace $c(t)$ using a sparse deconvolution algorithm with an $l_1$-norm constraint to enforce sparsity. This estimated spike train was then convolved with a smoothing window function $w(\Delta t)$ (e.g., a Gaussian or rectangular window) to produce a continuous estimate of the instantaneous firing rate f(t) in Hz:

$$f(t) = (\hat{s} \cdot w)(t) = \int \hat{s}(\tau) \cdot w(t - \tau) d\tau. \tag{S2}$$

This method provides a higher temporal resolution estimate of neural activity by effectively inverting the slow calcium dynamics, yielding a signal more suitable for driving rate-based network models.

## B    RECURRENT WEIGHT INITIALIZATION

The initial recurrent synaptic weights (section 3.4) were drawn from a truncated normal distribution,

$$w \sim \psi(\mu = 0, \sigma = \sigma_0, a = -2, b = 2), \tag{S3}$$

where the truncation interval $[-2, 2]$ ensures numerical stability by preventing extreme outliers. The standard deviation $\sigma\_0$ was set according to a variance-scaling principle widely used in deep learning to stabilize activity at initialization. Specifically, it was computed as

$$\sigma_0 = \frac{\sqrt{2/n_{\text{eff}}}}{\tilde{\sigma}_{\mathcal{N}}^{[-2,2]}}, \tag{S4}$$

where $\tilde{\sigma}_{\mathcal{N}}^{[-2,2]} = 0.87962566103423978$ is the standard-deviation correction factor for the truncated normal distribution on the interval $[-2, 2]$, and $n_{\text{eff}}$ denotes the effective number of synaptic inputs per neuron. The latter was estimated as the total number of non-zero entries in the structural adjacency matrix divided by two, reflecting the sparsity imposed by the connectome.

Importantly, only the non-zero synaptic weights specified by the connectome were initialized in this way and treated as trainable parameters, ensuring that optimization remained constrained by anatomical structure.

## C    LOW-RANK WEIGHT APPROXIMATION FOR RECURRENT CONNECTIVITY

We observed that training synaptic weights under connectome-constrained connectivity (Section 3.4) creates a severe memory bottleneck for BPTT (Section 4.1). This limitation arises because gradient computation requires storing a large number of intermediate states across both time and the densely connected recurrent graph, causing memory usage to grow prohibitively with sequence length. To address this challenge, we investigated a memory-efficient alternative based on a low-rank approximation of the recurrent connectivity matrix.

Specifically, the recurrent synaptic input is expressed as

$$\mathbf{I}^{\text{conn}}(t) = \mathbf{W}\mathbf{r}(t - 1) = \mathbf{L}\mathbf{R}\mathbf{r}(t - 1), \tag{S5}$$

where $\mathbf{W} \in \mathbb{R}^{N \times N}$ denotes the full recurrent weight matrix, $\mathbf{L} \in \mathbb{R}^{N \times k}$, $\mathbf{R} \in \mathbb{R}^{k \times N}$ are low-rank factors, and $N$ is the neuron number. During training, we set $k = 10$, which drastically reduces the number of parameters and the memory footprint while still enabling rich recurrent dynamics.

With this low-rank factorization, BPTT was able to successfully train the model on neural activity fitting tasks for short to moderate sequence lengths (Fig. S1B). Nevertheless, even with this reduction, scalability remained limited: at batch size 32, training failed with an out-of-memory error once the dataset length exceeded approximately 400 time steps.

## D    NEUROPIL FIRING RATE READOUT WITH A LINEAR TRANSFORMATION

In addition to the connectome-constrained readout mechanism (section 3.6), we also considered a more flexible approach in which neuropil activity is obtained through a learned linear transformation of neuron-level firing rates. Specifically, the firing rate of neuropil $k$ at time $t$, denoted $\mathrm{FR}^{\mathrm{neuropil}}k(t)$, is computed as

$$\mathrm{FR}_k^{\mathrm{neuropil}}(t) \;=\; \mathrm{ReLU}\left(\sum_j W_{k,j}^{\mathrm{out}}, r_j(t) + b_k\right), \tag{S6}$$

where $W_{k,j}^{\mathrm{out}}$ is a learnable weight mapping the firing rate of neuron $j$ onto neuropil $k$, and $b_k$ is a bias parameter. The ReLU activation enforces non-negativity of the predicted firing rate, consistent with experimental observations that calcium-imaging-derived neuropil signals are bounded below by zero.

This linear readout framework serves as an anatomically unconstrained baseline, in contrast to the connectome-based aggregation rule. The comparison between the two readout strategies highlights the role of anatomical priors in improving both accuracy and interpretability of whole-brain activity fitting.

## E    SUPERIORITY OF ONLINE LEARNING FOR TRAINING BASELINE MODEL

To establish a performance benchmark, a Gated Recurrent Unit (GRU) network (Chung et al., 2014) with 256 hidden units was implemented. The training of this baseline model revealed two critical advantages of the D-RTRL online learning method (Wang et al., 2024) over BPTT: convergence stability and memory efficiency.

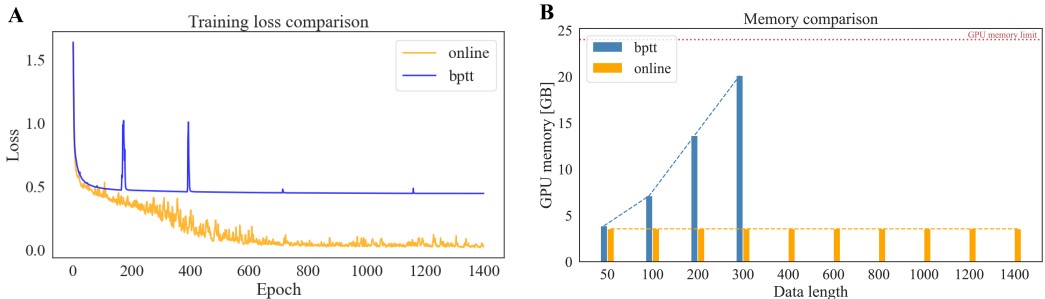

Figure S1:  **Comparison of BPTT and online learning in convergence and memory efficiency.** (A) Training loss of the GRU baseline model. When trained with BPTT (blue), the model failed to converge to a low-error solution, exhibiting unstable dynamics. In contrast, D-RTRL (orange) enabled stable convergence to a significantly lower final loss, demonstrating its efficacy for this task. (B) Memory usage of BPTT versus online learning. To evaluate GPU memory consumption, we compared both methods using low-rank connections (as BPTT is infeasible with the full connectome). BPTT memory usage grew rapidly with sequence length, exceeding typical GPU limits (dashed line) at a length of 400. The online learning method maintained substantially lower and sequence-length-independent memory consumption.

**Convergence stability:** Initial attempts to train the GRU using BPTT were unsuccessful, as the model failed to converge to a low-error solution (Fig. S1A). The BPTT loss curve exhibited high variance, reflecting the optimization challenges for this long-horizon task. Conversely, D-RTRL training resulted in stable convergence to a significantly lower final loss (Fig. S1A), enabling a fair architectural comparison.

**Memory efficiency:** Beyond convergence, BPTT is limited by its substantial memory footprint. As shown in Fig. S1B, BPTT memory consumption scales rapidly with sequence length, exceeding GPU capacity for long sequences (with low-rank connections). This constraint would be prohibitive

at the full connectome scale. In contrast, online learning maintains low, largely sequence-length-independent memory usage, making it practical for long neural recordings.

These results demonstrate that online learning (D-RTRL) offers both convergence stability and memory efficiency advantages over BPTT, justifying its use for training and comparison in this study.

## F    BASELINE MODEL COMPARISONS

To evaluate the efficacy of different model architectures in capturing neural dynamics, we compared our connectome-based model with a standard GRU network (256 hidden units). Both models were trained using the D-RTRL online learning algorithm and evaluated under the same protocol: for each test trial, the models were first warmed up for 250 time steps by feeding the ground-truth activity at each step, followed by 750 steps of autoregressive prediction where each step's input was the model's prediction from the previous step.

**A**

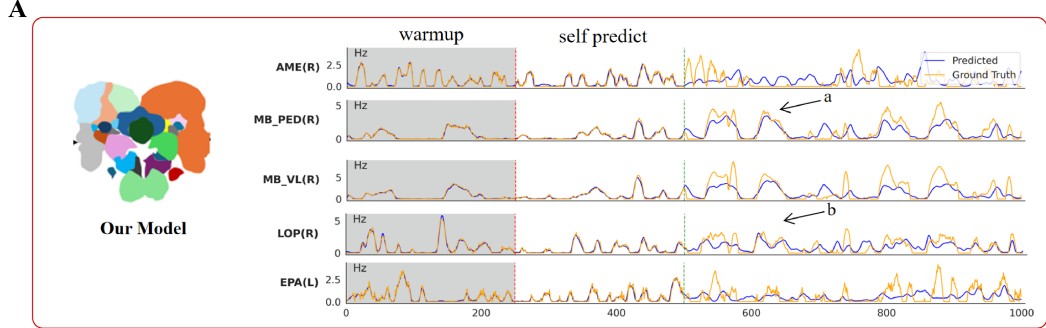

**B**

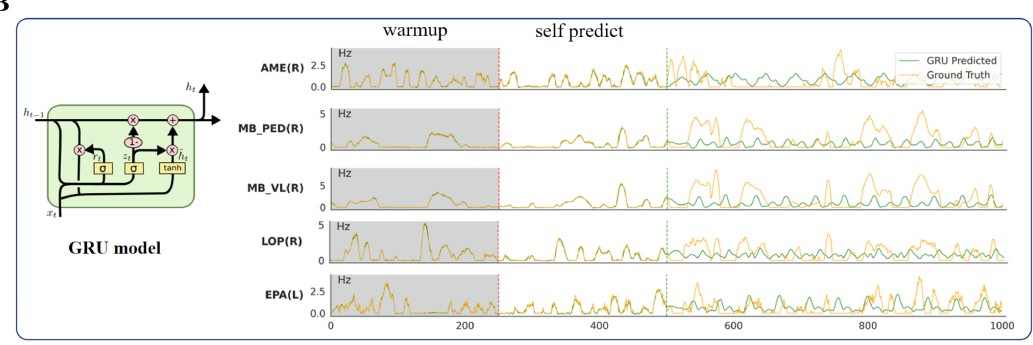

Figure S2: **Evaluation of connectome-based and artificial neural network models on neuropil activity.** (A) Connectome-based network model. The model successfully maintains diverse and realistic neural activity patterns throughout both the warmup and autonomous prediction phases, demonstrating robust capture of the underlying dynamics. (B) GRU network with 256 hidden units. The model's generalization is poor, particularly on unseen test data. it collapses into simplistic limit cycles during autonomous prediction on unseen test data, failing to sustain realistic dynamics. The vertical dashed lines demarcate the evaluation stages: the red line separates the 250-step warmup phase (left) from the 750-step autonomous prediction phase (right); the green line separates the first 500 steps (familiar training data) from the subsequent 500 steps (unseen test data).

**Qualitative analysis:** The GRU model exhibited poor generalization on unseen test data, frequently collapsing into simplistic limit cycles (Fig. S2B). In contrast, our model sustained rich and diverse activity patterns that more accurately mimicked the experimental data (Fig. S2A).

**Quantitative analysis:** The functional connectivity (FC) matrix derived from the GRU's predictions showed a low correlation with the empirical FC, with a Pearson correlation coefficient of $r = 0.211$ (Fig. S3). This performance was substantially lower than the correlation achieved by our model ($r = 0.556$). Furthermore, avalanche analysis revealed that the GRU's output failed to exhibit the

power-law scaling characteristic of neural criticality (Fig. S7A), indicating its inability to capture this fundamental dynamical property.

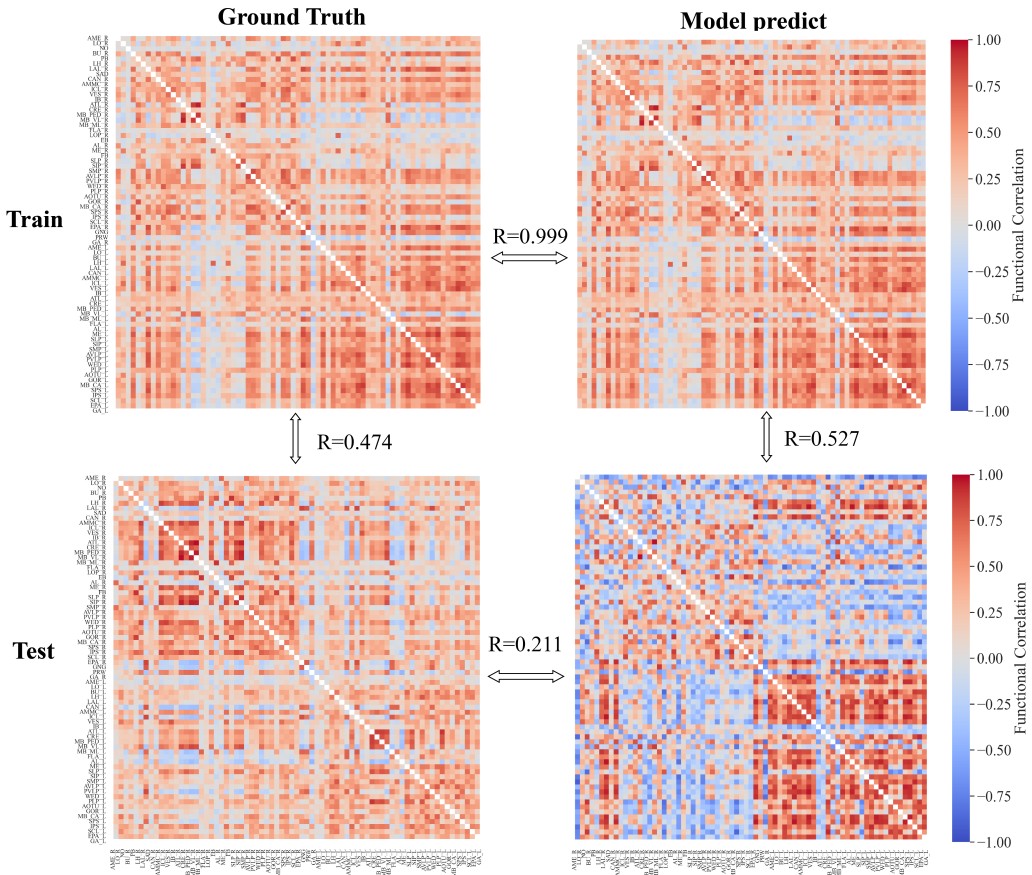

Figure S3: **Comparison between ground-truth and GRU-predicted neuropil functional connectivity.** The FC matrix derived from the GRU model's predictions shows low correlation with the empirical FC on test data($r = 0.211$), indicating poor accuracy in replicating the network-level correlations observed in the neuropil data.

## G    PARAMETER STABILITY AND REPRODUCIBILITY ANALYSIS

To assess the robustness and reproducibility of the learned model—key concerns for large-scale, high-capacity networks—we analyzed the stability of the final trained connectivity weights across multiple independent runs with different initializations.

**Systematic Shift Induced by Learning:** Comparing the initial uniform distribution (Fig. S4A, blue) with the trained distribution (orange) reveals a consistent, systematic shift. This indicates that the model purposefully adjusts parameters to capture underlying data structure rather than randomly memorizing training data.

**Cross-Initialization Consistency:** Models initialized from different prior distributions—a truncated normal distribution and a uniform distribution—converged to final weight profiles that were highly consistent across neuropil (Fig. S4B). The two profiles are statistically indistinguishable, indicating that the learned solution is not an artifact of a specific initialization scheme.

**Post-training consistency of neuropil weights:** Independent training runs starting from the same initial uniform distribution yielded virtually identical final weight vectors (Fig. S4C). This demonstrates the high determinism and numerical stability of the training process itself.

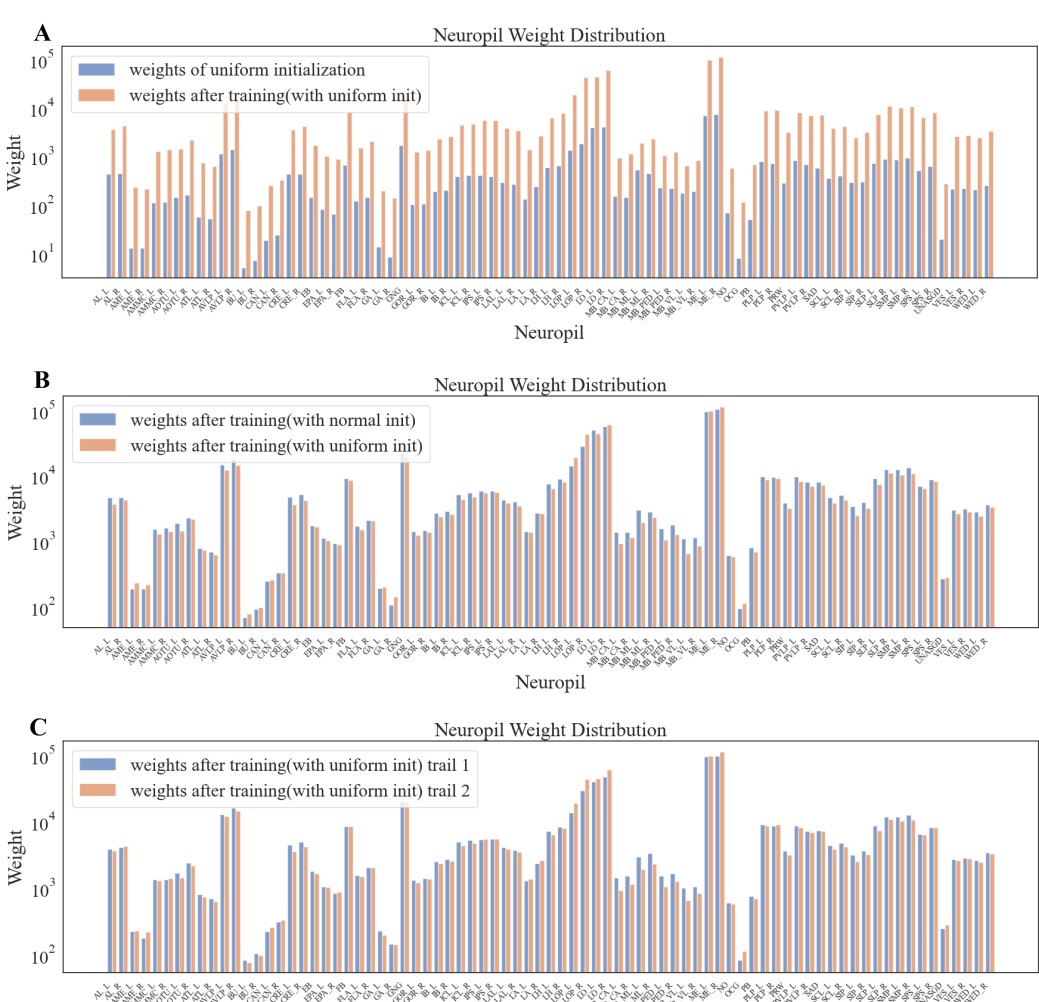

Figure S4: **Stability and reproducibility of learned connectivity weights under different initializations.** (A) Systematic shift induced by learning. Neuropil weights initialized from a uniform distribution (blue) are compared with the trained weights (orange). The consistent shift demonstrates that learning systematically adjusts parameters to capture data structure rather than random memorization. (B) Cross-initialization consistency. Final weights for each neuropil are shown for models initialized with a truncated normal distribution (blue) versus a uniform distribution (orange). Despite different starting points, the final weight profiles across brain regions are highly similar. (C) Post-training consistency of neuropil weights. Final weights from two independent training runs, both initialized from the same uniform distribution, are plotted. The near-perfect overlap demonstrates that the training process is highly deterministic and reproducible. In both panels, the x-axis represents different neuropils, and the y-axis represents the final learned neuropils weight values.

These results collectively demonstrate that our model robustly converges to a unique and stable parametric solution. This convergence is independent of the initial conditions, which effectively mitigates concerns of overfitting to idiosyncrasies of the training set and underscores the reliability of the parameters and the insights derived from them.

## H    DISTRIBUTION OF TIME CONSTANTS

The intrinsic time constant of individual neurons is a fundamental parameter governing their temporal dynamics. To understand how learning shapes the network's timescales, we analyzed the time-constant distribution across all single neurons in the model at three stages: initial (pre-training), trained (post-training), and the per-neuron change between them.

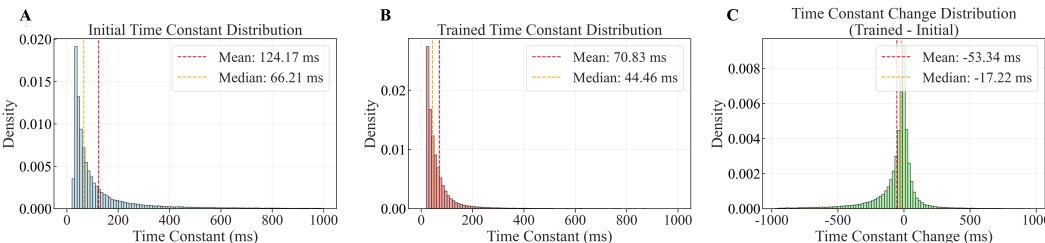

Figure S5: **Evolution of intrinsic time constants during learning.** (A) Initial distribution of time constants before training. The time constants are initialized within a biologically plausible range. (B) Distribution of time constants after model training. The learned time constants remain within a biologically plausible range, indicating that the model maintains realistic dynamical timescales. (C) Distribution of per-neuron changes in time constants (post-training minus pre-training). Most neurons exhibit relatively small changes, while a minority show substantial increases or decreases, indicating targeted adaptation of temporal processing in specific subsets of units.

Both the initial and learned time constant distributions (Fig. S5A, B) span ranges consistent with biological observations, confirming that the model's dynamics operate within physiologically relevant timescales. The distribution of per-unit changes (Fig. S5C) reveals that the majority of neurons underwent only modest adjustments, with a small subset exhibiting more pronounced shifts toward either longer or shorter time constants. This pattern suggests that learning selectively modulates the temporal dynamics of specific neuronal subpopulations, rather than uniformly shifting all units, potentially enabling the model to capture multi-timescale dynamics essential for generating realistic neural activity patterns.

## I    POST-TRAINING WEIGHTS EXHIBIT HEAVY-TAILED CHARACTERISTICS

The statistical distribution of synaptic weights is a fundamental characteristic of biological neural networks. We analyzed whether our model, after training, recovers a key statistical feature observed in the Drosophila connectome: a heavy-tailed distribution of connection strengths.

The empirical Drosophila connectome exhibits a heavy-tailed, approximately scale-free distribution of synaptic weights, as evidenced by the power-law fit to the tail of the absolute weight distribution (Fig. S6A). Remarkably, our model's weight distribution evolves toward a similar structure through learning. Starting from its initial state, the distribution of absolute model weights gradually develops a pronounced heavy tail as training loss decreases (Fig. S6B). This convergence suggests that the learning process not only optimizes for task performance but also implicitly drives the network's connectivity toward a statistical organization that mirrors a fundamental feature of biological brain networks.

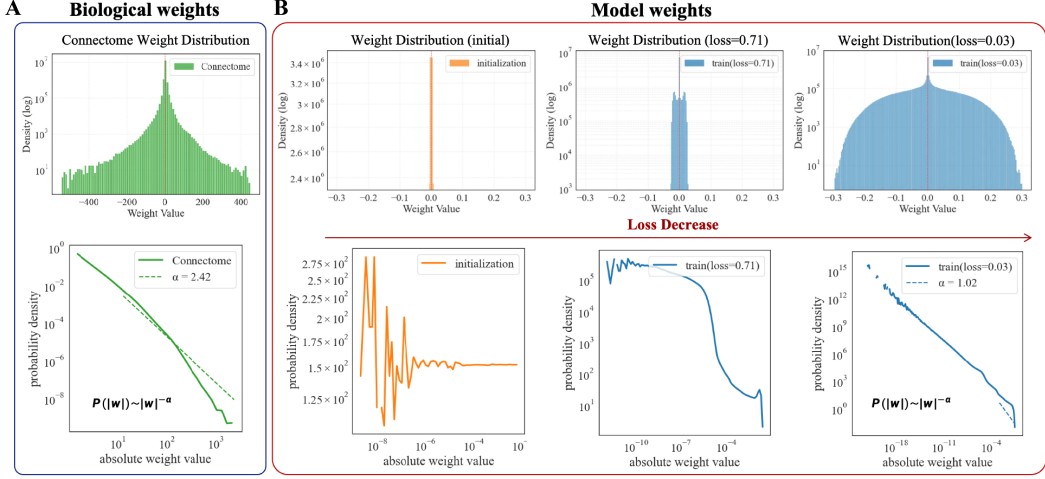

Figure S6: **Heavy-tailed weight distributions emerge during training** (A) Weight distribution of the Drosophila connectome. (Top) Histogram of connection synapse count. (Bottom) Probability density of the absolute connection weights. The tail region is well-fit by a power-law function (dashed line) indicating a heavy-tailed scale-free structure. (B) Evolution of the model's weight distribution during training. (Top) Histograms show the progression of the model weight distribution from initialization through different stages of training (with decreasing loss). (Bottom) The corresponding probability densities of the absolute weights. As training progresses and loss decreases, the distribution of absolute weights progressively develops a heavier tail and becomes better approximated by a power-law fit, converging toward a statistical structure resembling the biological connectome.

## J  AVALANCHE ANALYSIS OF NEURAL DYNAMICS

### J.1  SIGNAL PROCESSING AND AVALANCHE DETECTION

To analyze the critical dynamics of neural activity, time series were binarized and neuronal avalanches were detected using established methods.

**Signal binarization:**  Activity traces $x_i(t)$ for each unit (neuron or neuropil) were converted to binary events $b_i(t)$ by applying a unit-specific threshold:

$$b_i(t) = \begin{cases} 1, & \text{if } x_i(t) > 3\sigma_i, \\ 0, & \text{otherwise,} \end{cases}$$

where the baseline noise level $\sigma_i$ was robustly estimated from the median absolute deviation (MAD) of the signal.

**Avalanche detection:**  Avalanches were defined as contiguous periods of global activity, where at least one unit was active ($A(t) = \bigvee_i b_i(t) = 1$). For each avalanche, its duration $D$ (in time steps) was recorded.

**Power-law fitting:**  To assess scale-free criticality, we tested whether the avalanche duration distribution $P(D)$ followed a power law, $P(D) \propto D^{-\alpha}$. The scaling exponent $\alpha$ was estimated via linear regression in log-log space.

### J.2  AVALANCHE DURATION DISTRIBUTIONS REVEAL DIVERGENT DYNAMICS

The analysis of avalanche durations highlights a fundamental difference in the dynamical regimes captured by the GRU baseline and our model.

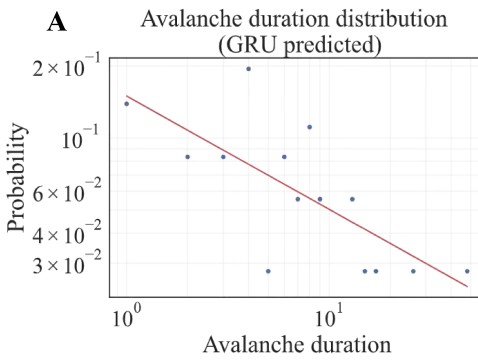 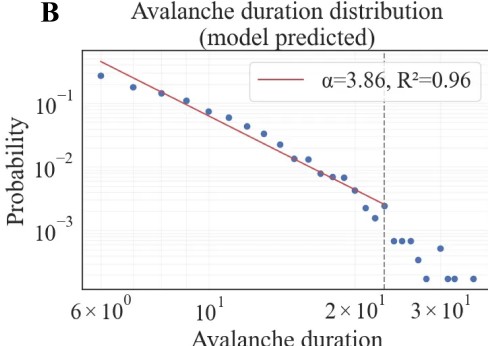

Figure S7: **Avalanche duration distributions of the GRU baseline and our model** (A) GRU model. The distribution of avalanche durations does not follow a power law, indicating a lack of critical, scale-free dynamics. (B) Our model (early training). The distribution is well-fit by a power law with an exponent $\alpha \approx 3.86$ (red line). This high exponent may suggest a supercritical dynamical regime.

As shown in Fig. S7A, the avalanche durations generated by the GRU baseline do not adhere to a power-law distribution, failing to capture the scale-free signature of neural criticality. In contrast, our model, even in early training stages, produces avalanches whose duration distribution is consistent with a power law (Fig. S7B). The fitted exponent of $\alpha \approx 3.86$ is notably higher than typical critical exponents, potentially indicating that the network operates in a supercritical regime characterized by amplified and prolonged activity cascades. This divergence underscores our model's enhanced capability to generate biologically plausible, collective neural dynamics compared to the standard RNN baseline.

## K  APPLYING SUGAR SENSORY STIMULATION TO THE MODEL

Our model architecture and training framework are fully compatible with fitting stimulus-driven or task-based neural activity. The key distinction from modeling resting-state activity lies in the introduction of additional parameters to represent external stimuli, while the core training paradigm remains unchanged.

Furthermore, a model trained on resting-state data can be directly used to probe its dynamical response to external inputs. To illustrate this, we conducted a simulation in which a simulated sugar stimulus was applied to the left gustatory receptor neurons within the model, analogous to experimental paradigms used to study taste processing (Shiu et al., 2024). The consequent activity changes in downstream mouthpart motor neurons (MN9) were then monitored and analyzed.

The model successfully captured a basic sensorimotor transformation: activation of the appetitive taste pathway led to excitation of the motor neurons. A key observation was a pronounced lateral asymmetry in the motor response. Following unilateral stimulation on the left side, the contralateral (right) MN9 neuron exhibited a markedly stronger increase in firing rate compared to the ipsilateral (left) MN9 neuron, as detailed in Table S1.

Table S1: Motor neuron response to simulated sugar stimulation

| cell type | side | firing rate (no stimulus) | firing rate (with stimulus) | increment | magnification |
|---|---|---|---|---|---|
| MN9 | Right | 0.10054 | 5.35326 | 5.25272 | 53.24 |
| MN9 | Left | 0.09425 | 2.23371 | 2.13946 | 23.70 |

The **increment** is defined as the difference between the average firing rate during the stimulus period and the average firing rate during the baseline (no stimulus) period. The **magnification** is the ratio of the average firing rate with stimulus to the average firing rate without stimulus. This demonstration confirms that the resting-state-trained model retains a functionally structured input-output map,

capable of generating biologically plausible, stimulus-specific motor dynamics under connectome constraints.

## L    ADDITIONAL SUPPLEMENTARY FIGURES

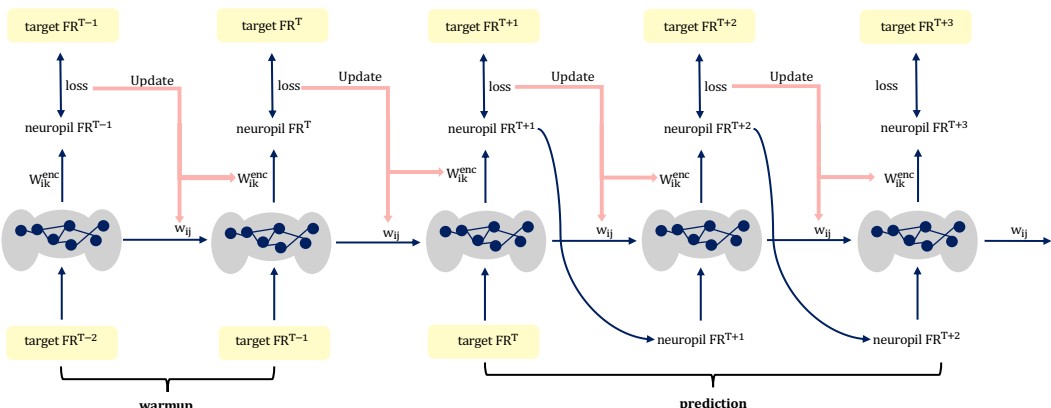

Figure S8: **Two-stage training pipeline.** The model was trained using an online learning framework consisting of a warm-up phase and a prediction phase. This framework updates synaptic weights with a combination of instantaneous loss and eligibility traces, thereby avoiding the high memory cost of backpropagation through time. In the warm-up phase, the model received the actual neuropil firing rate from the previous time step to predict the current rate. The resulting prediction error was used to compute gradients, which were accumulated over multiple steps before each weight update. In the prediction phase, the model generated autonomous activity: initialized from experimental data, it then used its own predicted output from the previous step as input. At each time step, the prediction was compared against the target sequence, training the model to sustain its dynamics independently. The online framework enabled efficient and scalable weight updates throughout this recursive process.

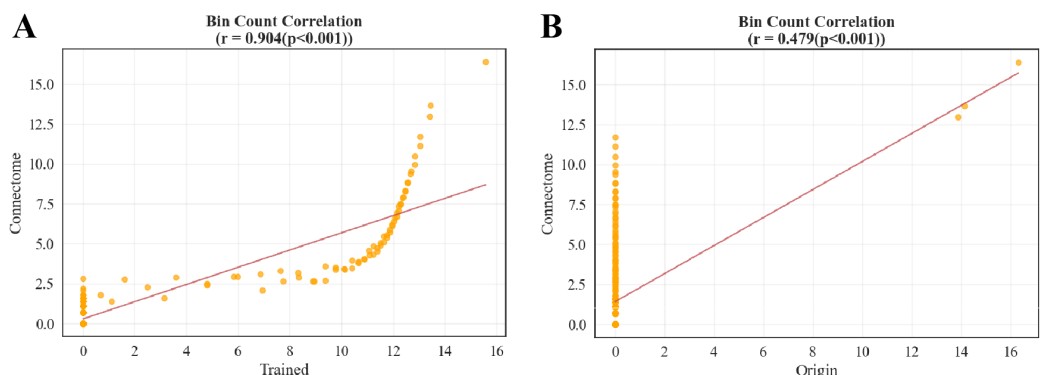

Figure S9: **Comparison of bin-count correlations within connectivity groups.** (A) Correlation between the trained model's recurrent weight distribution and the empirical connectome weight distribution. The scatter plot shows the correspondence of frequency counts across identical value-range bins, with both axes representing log-transformed frequencies ($\log(\text{bin count} + 1)$). The red dashed line indicates the linear regression fit, yielding a Pearson correlation of $R = 0.904$, demonstrating a strong alignment between the trained and biological distributions. (B) Correlation between the untrained weight distribution and the connectome distribution. The red dashed regression fit shows a Pearson correlation of $R = 0.479$, indicating only modest similarity prior to training and highlighting the emergence of biologically consistent weight patterns through optimization.

