# OpenReview forum: "Online Fitting Connectome-constrained Drosophila Whole-brain Model Reproduces Critical Resting-state Dynamics"
_ICLR.cc/2026/Conference — Submitted to ICLR 2026_

### Official Review · Reviewer_2GYa · 2025-10-28

**Soundness:** 2
**Presentation:** 2
**Contribution:** 2
**Rating:** 2
**Confidence:** 4

**Summary:**

This paper tackles memory-efficient parameter estimation in biologically plausible models of Drosophila (but really neural models more generally) neural networks.  The paper makes the observation that BPTT is not feasible for such large networks, and so applies a recent online method to reduce the memory footprint (at the expense of biasing gradients).  Some examination of the learned parameters are presented.

**Strengths:**

1. I think this is a super promising line of work.  Developing memory- and work-efficient algorithms for fitting neural models is super exciting, because it allows us to study the variability in the underlying model between specimens, before/after specimen training etc.  If we could do this efficiently, then the model itself also becomes a learnable "parameter", which allows model discovery to come to the fore.
2. Actually building a mega-scale model of an entire connectome and hooking it up to an inference and learning algorithm is no small undertaking. It's great to aim big and try and fit the whole thing; so many papers would try and fit smaller-scale models. Kudos to the authors.
3. The authors try and examine and pull apart the results with some depth and breadth, which can/could lead to some interesting insights down the road.

**Weaknesses:**

# Summary
Ultimately, I do not feel like this paper is ready for publication.  This paper falls into an awkward position:  it is a (from a methodological angle, at least) fairly vanilla application of an existing methodology, to a model that predominantly exists; some new model components are presented, but there is no empirical evaluation of their validity.  In contrast, the paper angles itself as tackling one of the biggest open problems in neuroscience, and so there is quite a high bar for evidence for that.  While it is a promising start, the empirical results aren't as convincing as I would need to see to for the claimed "science", and the methodological/modeling innovations are lacking in a pure machine learning sense.  I really encourage the authors to continue this line of work though, because if you're successful this would be an _enormous_ breakthrough.  Good luck


# Weaknesses (roughly in descending order of significance)
1. The experimental evidence is weak.  It seems like the evidence hinges on Figure 3 (see my comments below).  I am not convinced a good generative model has been recovered, and the correlation plots are a fairly blunt instrument for extracting neuroscientific understanding.  From a methodological standpoint, I'd like to see much more analysis of the bias and variance of estimators, how these scale with data length and across methods/approximations, how accurately parameters are recovered for various synthetic network sizes etc.  On the neuroscience side, I'd love a thorough examination of some functional assemblies:  if we can't say "this parameter is interesting", then what even is the point of biophysically accurate models!  Comparing distributions of learned parameters to hypothetical parameters is not especially convincing either.  For instance, Lappalainen study activations under different inputs, where the expected activation pattern is known but not trained against.  This is the sort of super compelling neuroscientific evidence and insight that is required to really make this a compelling neuroscience contribution.

2. On the experiments themselves:  it seems like most of the experiments are in relation to 500 time steps of training data for 73 neuropil regions?  This is a total of 36,500 observations (a lot of which are highly correlated).  If I understand correctly, your model has over 100k states and 15M connections.  Therefore, this model is unbelievably over-parameterized, even with the connectome constraints.  I am therefore _very_ dubious as to any insights drawn from the model, because this model could, essentially, fit any smooth system in a multitude of ways.  For instance, I would love to see a correlation analysis of learned parameters across different random initializations.  If the learned parameters are highly variable, then that is evidence that little reproducible scientific insight is being generated (despite getting good training reconstructions).

3. There is some evidence for this in Figure 3, in fact, where the training reconstructions are perfect; but the test reconstructions are comparatively very poor.  This suggests to me that the training set is being overfit to, and results in less-than-ideal test predictions.  You can get good correlation by accurately predicting the quiescent point.  I'd like to see a self-normalized or cumulative distribution normalized analyses to really look into how accurately, in a more robust probabilistic/calibration sense, the predictions are.  See my question below, I don't understand the role or implementation of $F^{neuropil}$, but is there a chance that is driving the test reconstructions, and not the dynamics?  I'd like to see an ablation of that.

4. The neural model you present is quite low fidelity -- it is essentially a single-compartment, net-excitatory-only, non-linear model of calcium dynamics.  Calcium dynamics are not the driver neural dynamics, and also act as a low-pass filter to potential dynamics.  As a result I question whether "correctly" fitting such a mechanistic model that is over simplified would ever lead to really meaningful neuroscientific insight.  This is the sort of really top-end insight that is needed when your paper sets out to do real science.

5. The explanation of the model and inference methods are quite unclear (in my opinion).  I think these sections would benefit heavily from overhauling and improving the writing.  There is hardly any introduction or discussion of the strengths, weaknesses or applicability of DTRL. The exposition around the firing rate resolution (Fig 1D), as well as eq. (3), are especially unclear.  I'd love to really understand the neurobiological underpinnings of these models and why they are so useful, but the exposition, unfortunately, is exceptionally poor.  It is really difficult to get excited for a model where there are large parts of it I simply cannot parse out.

6. The paper takes quite a long time to get going, with notation, parameters equations etc spread over several pages.  IMO the paper could really benefit from a short "Problem Setup"-esque section that quickly, compactly and concretely defines the problem setup early on.  This would include parameters and their spaces;  transition and emission functions with their arguments and outputs etc.

7. I get that this is essentially an application of D-RTRL to Drosophila data, but I would still like a more thorough examination (or evidence) of BPTTs non-applicability.  For instance, Aicher+ [UAI, 2020] explore truncating BPTT, trading off bias with cost.  There is some exposition on the difference between BPTT and D-RTRL in Section 4.1, but it's very short and not super informative.  Right now I would not be convinced into using your method for this problem.  Examination of this wider literature and really zooming in on the applicability of different methods (with experiments) would really shore up your methodological footing.

8. I am just about willing to believe Figure 5a, but that first and last point are doing a lot of work for that correlation/LoBF, some kind of sensitivity analysis/random repeats would be nice here just to ensure results are not buoyed by outliers.  Similarly, I do not find the avalanche results particularly compelling.  None of the result actually seem to follow a power law, and even if they did, I think its weak evidence without a lot more exposition on how these measurements were obtained/real-world validation/ablations etc.  i.e. Are these phenomena just a feature of networks generally?  What model components are critical to generate this effect?

9. A batch size of 16 is huge in some domains.  Sometimes LLMs (for instance) use a batch size of one!

10. Figure 4c is interesting, but why do we believe they should be aligned?  The whole hypothesis of this paper is that we can't know the functional weight ahead of time, and so the more right the hypothesis is, the worse this result becomes.  I, admittedly, do not know how to remedy this, but it is a difficult wrinkle in the analysis.

11. I don't think some of the claims are borne out:  e.g. the y=x reference line claim in 433.

**Questions:**

Q1. Does $f$ being ReLU mean that there can be no _net_ inhibition?  I.e., individual neurons can inhibit as much as other neurons have excited?

Q2. Are the observations really at 1.2**Hz** (cf. Line 214).  This seems incredibly low to me, to the point where I would be amazed that we can resolve _any_ neural dynamics!

Q3. What is the low-rank approximation in Section 4.1?  This experiment needs much more introduction and discussion.

Q4. Were multiple random seeds used?

---

> ### Author Response · Authors · 2025-12-02
>
> Thank you very much for highlighting both the value and the concerns.
>
> We need to clarify that the value of this model lies in its being the first neural dynamic model that operates at single-neuron resolution, is constrained by the whole-brain connectome structure, and fits functional data. Methodologically, this study is the first to overcome the computational bottleneck in building large-scale, high-biological-fidelity whole-brain models, establishing a new paradigm for studying structure-function relationships within a unified framework—it is not merely a routine application of existing methods. Regarding the model's effectiveness you mentioned, we have provided a clear explanation in the appendix.
>
> You are absolutely right. The relationship between structure and function is indeed a key open question in neuroscience. The current aim of this study is to provide such a whole-brain modeling method as a new approach and paradigm for investigating structure-function relationships. Many interesting hypotheses, including how structural connectivity shapes dynamics, could not be presented in this paper due to space constraints. In the future, we will leverage this functional, connectome-constrained, and interpretable model to address these critical questions.
>
> We thank you for your attention and suggestions.
>
> **Response to W1 & W3:**
>
> We thank the reviewer for the questions regarding model performance. Please refer to our response in the “Model performance and baseline evaluation” section.
>
> **Response to W2:**
>
> We thank the reviewer for raising this critical point. we focus on the stability of learned parameters across different random initializations. Our results strongly demonstrate that, despite the large model capacity, the learning process is highly robust and reproducible.
>
> 1. **Convergence Consistency Across Initializations** (Figure S4B, Appendix G): We compared the final learned weight distributions starting from vastly different initial distributions (uniform vs. truncated normal). The results show that they converge to **highly similar distributions**, proving that the final state is independent of initial random conditions.
> 2. **Post-training consistency of neuropil weights** (Figure S4C, Appendix G): Two independent training runs starting from the same uniform initialization yield **highly consistent** final weight distributions. This demonstrates the stability and determinism of the training process itself.
> 3. **Systematic Shift Induced by Learning** (Figure S4A, Appendix G): Comparing the initial distribution with the trained one reveals a **systematic and consistent shift** caused by learning. This indicates that the model is not merely memorizing the data randomly but is purposefully adjusting parameters to capture the underlying structure of the data.
>
> In summary, these analyses demonstrate that our model is not a chaotic system that can “fit any smooth system in a multitude of ways.” Instead, it robustly converges from different starting points to a **unique and stable parametric distribution solution**. This reproducibility and insensitivity to initial conditions form the foundation for deriving reliable scientific insights from the model.
>
> **Response to W4 & Q1:**
>
> Our model is an EI-constrained network based on the synaptic polarity from the real connectome (see original text: line 246). Polarity was assigned based on neurotransmitter identity: acetylcholine and dopamine were considered excitatory (+1), while GABA, glutamate, octopamine, and serotonin were considered inhibitory (-1), consistent with their dominant physiological effects in the fly brain.
>
> **Response to W5:**
>
> We thank the reviewer for this question. The advantages of DTRL are clearly outlined in (Wang et al., 2024), including its model universality, computational efficiency, and engineering usability. In our practical implementation, DTRL also demonstrated excellent performance. For instance, when training a GRU model on the task presented in this paper, DTRL outperformed BPTT, which exhibited poor performance even after full convergence (see Appendix E and Figure S1).
>
> **Response to W6:**
>
> We thank you for this suggestion. We will revise the relevant section in the manuscript to make it more concise and accessible.
>
> **Response to W7:**
>
> Thank you for raising this point. We should compare the impact of different BPTT variants on model training performance. However, in practice, even with a data length of 1, BPTT could not be run on an 80GB A100 GPU. Therefore, a comparison among various truncated BPTT variants is not feasible.

---

> > ### Author Response · Authors · 2025-12-02
> >
> > **Response to W8:**
> >
> > Thank you for this question. Figure 5a displays avalanche statistics derived from real *Drosophila* neuropil calcium imaging, which is an empirical observation and thus not a product of random repetition. The heavy tail in Figure 5b is a known finite-size effect common in such systems. Following standard practice in the field, we excluded this tail region during the power-law fitting.
> >
> > Regarding the concern that this might be a common network artifact, we performed two control experiments. First, a well-trained GRU model on our task produced avalanche dynamics that do not follow a power law (see figure below). Second, early in our model’s training (before full loss convergence), the avalanche distribution did follow a power law, but with a very high exponent (α=3.86), indicating near-supercritical network dynamics. This demonstrates that the critical dynamics in our final model are a learned property, not a universal feature of all networks.
> >
> > **Response to W9:**
> >
> > We thank the reviewer for noting the batch size setting. We need to clarify that our use of a batch size of 16, rather than 1, is primarily an engineering optimization decision to fully utilize GPU parallel computing capabilities, not a limitation of the algorithm itself.
> >
> > Modern GPUs have highly parallel architectures designed to process multiple data samples simultaneously. Using a larger batch size can significantly improve GPU utilization and computational throughput. For a given dataset size, a larger batch size reduces the total wall-clock time required for training. For our large-scale model with 138,639 neurons, the forward pass and gradient computation for a single sample already involve massive matrix operations. Packing multiple samples into a batch for parallel processing effectively amortizes memory access overhead and kernel launch latency, thereby accelerating the training process. This is standard practice in deep learning.
> >
> > Importantly, our online learning framework fully supports, in principle, a batch size of 1—i.e., true per-sample or per-timestep updates. The eligibility trace updates in the D-RTRL algorithm (Section 3.7, Eqs. 5-10) are performed independently for each timestep, and parameter updates can be executed at any point. A batch size of 16 means we compute gradients in parallel over 16 independent trials and then average these gradients for a parameter update. This is algorithmically equivalent to a batch size of 1 (updating per trial), but the former improves computational efficiency through batching.
> >
> > **Response to W10:**
> >
> > You are absolutely right. In fact, what we aim to highlight is that as the loss decreases, the model weights gradually develop this heavy-tailed distribution characteristic—i.e., many small weights and a few large ones. This heavy-tailed distribution is a core feature of most biological systems, including the connection counts in the EM-reconstructed connectome (see Appendix I and Figure S6).
> >
> > **Response to W11:**
> >
> > We thank the reviewer for raising this issue. We will address this by revising the relevant section in the manuscript.
> >
> > **Response to Q2:**
> >
> > You are correct that the sampling rate for the calcium activity was indeed 1.2 Hz.
> >
> > **Response to Q3:**
> >
> > The low-rank approximation assumes, in the case of fully connected recurrent connections, decomposing the full-rank weight matrix W into two smaller matrices L and R to reduce computational and memory costs. This is a standard practice in machine learning.
> >
> > **Response to Q4:**
> >
> > Thank you for raising this important question. Yes, we rigorously assessed the robustness of our model’s results by using multiple random seeds.

---

### Official Review · Reviewer_4rxR · 2025-10-29

**Soundness:** 2
**Presentation:** 3
**Contribution:** 2
**Rating:** 2
**Confidence:** 4

**Summary:**

In this paper, the authors fit a connectome-constrained recurrent model of the drosophilia brain with single-neuron resolution. To avoid prohibitively large memory demands of BPTT, they perform this fitting using D-RTRL, an approximation that allows for memory-efficient parameter updating. They argue that their fitted model yields decent reconstruction of dynamics and functional connectivity at the level of neuropils. Further, they argue that their fitted weights reconstruct summary statistics and weight distribution properties of those found in the actual connectome. Finally, they argue that their fitted model reproduces observed criticality phenomena such as neuronal avalanches.

**Strengths:**

- The writing is reasonably clear, and the model formulation is easy to understand.
- To the best of my knowledge, this paper presents the first connectome-constrained recurrent model of drosophilia neural dynamics that is fitted at the individual neuron level and the whole-brain scale.

**Weaknesses:**

- The claim that the fitted model actually significantly reproduces drosophilia neural dynamics is quite unconvincing (Fig. 3). While it is indeed a tall task to actually predict neural activity well many timesteps into the future, the test-time predictions seem shockingly poor, especially for a connectome-informed model. The paper's claim that the model captures the true dynamics beyond the train set is vastly overstated, as segments labeled "a" and "b" feel cherry-picked. Summary statistics like functional connectivity, on the other hand, are much easier to estimate, and don't even require a dynamical model to obtain a reasonable estimate, so I find it only mildly impressive that the model slightly improves over the baseline of using the train-time functional connectivity estimate.
- The implicit claim made by this paper is that the inclusion of detailed connectome information at the level of individual synapses and neurons yields a model that would otherwise not be able to reproduce various macroscopic properties of true drosophilia neural dynamics. Setting aside my concern above that the fitted model does not achieve this convincingly, the evidence that connectome constraints are playing a large role in shaping the properties of the final fitted model is also weak. In particular, there are no null models that are compared to. There are many sensible null models that one could use that should be equally easy to train: randomizing $\text{sgn}_{ij}$, randomizing the overall connectivity graph imposed, partial randomizations that preserve in-degree, out-degree, total number of outgoing excitatory synapses, etc. Also, a simple RNN with units at the level of neuropils feels like a very accessible baseline. I strongly suspect that at least one of these null models could reproduce many of the macroscopic findings like the final weight distribution, moderate functional connectivity estimation gains, and criticality.
- Limitations of the approach do not seem to be acknowledged.

**Questions:**

- If graded connectome weights (as opposed to binary) are available, as suggested by Fig. 4, why are they not used in any way? If not for direct constraints on the weights, at least for initializations?

---

> ### Author Response · Authors · 2025-12-02
>
> We thank you for reviewing our manuscript and providing valuable feedback. We have carefully considered each of your comments and respond to them point by point below. For clarity, we address several related core issues in the section titled "Model performance and baseline evaluation," which will be referenced in the subsequent specific responses.
>
>
> **Response to W1 & W2:**
>
> We thank the reviewer for raising this important question. Please refer to our response in the “Model performance and baseline evaluation” section.
>
>
> **Response to W3:**
>
> We thank the reviewer for pointing out this important issue. We acknowledge that the current version of the manuscript does not sufficiently discuss the limitations of the study. We will add a dedicated "Limitations and Future Directions" section in the revised manuscript to systematically outline the main limitations of this work, including the following aspects: limitations arising from model simplifications, biological uncertainty of the background input component, mismatches in data resolution, limitations of the training data, approximations in the online learning algorithm, and insufficient understanding of the mechanisms underlying emergent criticality.
>
>
>
> **Response to Q1:**
> We did attempt to initialize the model weights directly using the synaptic counts from the connectome (i.e., synaptic strengths estimated from EM data). However, this initialization strategy resulted in poor training performance, with the model struggling to converge to a reasonable loss level. This outcome is not surprising, as in deep learning and neural network training, the initialization scheme critically impacts the optimization process. Inappropriate initialization can lead to vanishing/exploding gradients or getting stuck in poor local optima. While the synaptic counts provided by the EM connectome reflect anatomical connection strengths, the scale and distribution of these raw values may differ significantly from the effective synaptic weights required to support specific dynamical states (e.g., resting-state oscillations). Furthermore, EM data cannot capture synaptic functional heterogeneity, short-term plasticity states, or modulatory effects, all of which can influence effective synaptic transmission strength.
>
> Therefore, we adopted the widely validated variance scaling random initialization strategy from the deep learning field (see Appendix B for details). This initialization method ensures stability of network activity in the early stages of training, avoids numerical instability, and provides a good starting point for gradient descent optimization. Crucially, although our initial weights are random, the connectome topology (i.e., which neurons are connected) and synaptic polarity (excitatory or inhibitory) are always strictly constrained by the FlyWire connectome. In other words, we fix the "existence of connections" and the "sign of connections," while treating the "strength of connections" as free parameters optimized in a data-driven manner.
>
> It is worth emphasizing that, despite random initialization, the distribution of trained synaptic weights exhibits statistical properties highly consistent with the experimental connectome. As shown in Figure 4 and Appendix I, the optimized weights form a heavy-tailed distribution where most connections are weak and a few are very strong, closely matching the pattern observed in the FlyWire connectome's synaptic count distribution. This result indicates that, although we did not explicitly optimize towards EM synaptic strengths as a target, the data-driven optimization process spontaneously pushes the weight distribution towards an organizational pattern consistent with biological observations.

---

### Official Review · Reviewer_MzjY · 2025-11-01

**Soundness:** 2
**Presentation:** 3
**Contribution:** 1
**Rating:** 2
**Confidence:** 4

**Summary:**

The paper proposes a firing rate network model based on the FlyWire drosophila connectome with learnable weights and time constants, but fixed synaptic polarity. The authors note that BPTT memory usage scales as O(N * t) and instead use the D-RTRL method implemented within the BrainScale framework, which scales as O(N). The model is trained with per-neuropil calcium imaging data acquired in a resting state at 1.2 Hz and for 73 neuropils. Post-training evaluations show biologically plausible behavior of the trained network.

**Strengths:**

- The paper is clearly written, easy to read, and cites relevant prior work.
- Building scalable connectome-constrained models is an important area of research.
- Evaluation of the trained model includes multiple aspects (functional connectivity, synaptic weight distributions, and criticality).
- The paper includes source code.

**Weaknesses:**

- Limited novelty, and diversity of the study and evaluation. The paper applies an existing algorithm to a new dataset and shows that multiple high level properties (avalanche statistics, synapse strength distribution, neuropil-level functional connectivity) have reasonable values. To make a strong statement about the applicability of D-RTRL to this type of models, ideally the authors would also compare to prior results obtained with BPTT, such as those of (Chen et al., 2022) or (Lappalainen et al., 2024).
- The model includes a direct and learnable input from every neuropil to every neuron (Eq. 3), in addition to the inputs driven by the connectome. This is not biologically realistic and not justified well. The paper should explain this better, explicitly include the parameter count with and without this term, and include an ablation where this term is dropped, as well as analysis of resulting weights where the term is kept (for instance, how often does it happen that a neuron couples to neuropils with no existing synaptic connectivity?).
- Limited control experiments. The authors attempt training their model with a linear readout instead of the standard connectome-weighted one (Eq. 4). Ideally one would also see experiments where the weight matrix itself (in I_conn) is perturbed to show that the real connectivity matters.

**Questions:**

- line 135: "BPTT, while effective, remains biologically implausible (Lillicrap et al., 2020)". This sentence seems out of place. Biological implausibility is not a reason why the modeling approach cannot be extended to the whole brain scale.
- line 427: "where synaptic spine counts ...": very few neurons in the fly brain have spines. Was the term "spine" intended here?
- line 259: "numerous sources of input remain unaccounted for during resting states": what are those sources?
- Fig. S5 is truncated
- Fig. 2B shows that loss is lower with BPTT; does that matter?
- Please report the timestep in the main text; otherwise "biologically realistic" timescales are hard to interpret.
- The deconvolution model used assumes spiking neurons; but not all neurons of the fly are spiking; how is this accounted for?
- Compared to the real connectome distribution, there are too many strong connections. Can you comment on why that might be the case?
- How much memory would be needed to train the model with BPTT? Could you use multiple GPUs and compare the results?
- Is there any data showing that training over longer horizons helps? How could that be quantified? Fig. 2B does not show an obvious gain with longer time horizons.
- Isn't the R=0.75 of predicted train/test activity, which is higher than R=0.474 for real data, a sign of overfitting rather than "robust temporal consistency"?
- Are the trained time constants biologically plausible?
- Synaptic strengths can be estimated from EM. Could you please include an ablation in which they are kept fixed and a comparison between the actual strengths and the post-optimization ones in your study, computed at the synapse level? (e.g. a distribution of the real-trained weight difference).
- The paper uses 138k neurons, but measures only the averaged outputs for 73 neuropils. Could one build a much simpler and lower-dimensional model that only uses mesoscale neuropil-level activity instead of tracking individual neurons? How would that impact the results?

---

> ### Author Response · Authors · 2025-12-02
>
> **Response to W1:**
>
> Thank you for the question. Our study differs fundamentally from the mentioned works (Chen et al., 2022; Lappalainen, 2024) in its objectives, resource demands, and algorithmic comparisons:
>
> **1.Different Research Objectives:** Chen et al. and Lappalainen et al. train models constrained by local brain region connectomes on specific functional tasks (e.g., image classification, optical flow prediction). Our model aims to **directly predict real, spontaneous neural activity**. Training for functional tasks does not focus on precise neural activity reproduction but rather on statistical comparisons with real neural activity.
>
> **2.Orders of Magnitude Difference in Resource** Chen et al. use the BPTT method for training. Training a model with 50,000 neurons requires **160 A100 GPUs**. In contrast, this study uses the D-RTRL online learning method to train a network of **nearly 140,000 neurons** with only **1 GPU**, which is impossible using the BPTT method. Overall, our method establishes a resource-optimized, online learning-based paradigm for neural activity prediction.
>
> **Response to W2 & Q3:**
>
> The modeling of background input is indeed artificial and not biologically plausible. Neuronal activity during resting states is influenced by numerous unmodeled input sources, including: diffuse neuromodulatory systems, non-specific background activity, residual effects of sensory input, and modulatory effects of glial cells. Therefore, we model these factors as an artificial input component.
>
> The number of parameters for the input encoding weights is approximately 138,639 x 73 = 10,120,647; the number of parameters for recurrent connections is approximately 15,091,982. This term indeed increases the parameter count by about 67%, but it remains extremely sparse and efficient compared to a fully connected network (138k × 138k ≈ 19 billion parameters).
>
> We will fully consider your suggestion in future work to conduct various ablation experiments on the input.
>
> **Response to W3:**
>
> Thank you for raising this point. In future work, we will explore the differences in training performance when the connectome is included or excluded from the trained connections. However, an obvious distinction is that the connectome-constrained model holds a significant advantage in interpretability, enabling its use for virtual neuroscience experiments and controlled analysis.
>
> **Response to Q1:**
>
> We thank the reviewer for pointing this out. As suggested, we will revise the text on line 135 to: “Scaling BPTT for whole-brain-scale modeling is exceedingly difficult due to its massive computational resource demands.”
>
> **Response to Q2:**
>
> We sincerely thank the reviewer for this precise correction. We fully agree that the term "spine" is inaccurate in the context of Drosophila. We will revise the statement on line 427 to: “where synaptic connection counts…”, to ensure scientific accuracy in terminology. This wording is consistent with the terminology used in the FlyWire connectome data.
>
> **Response to Q4:**
>
> Thank you for pointing out this issue, we will correct it.
>
> **Response to Q5:**
>
> To clarify, applying BPTT to a network of our target scale (~140k neurons) is infeasible, as even a small training batch would exceed GPU memory limits. In contrast, D-RTRL can operate at this scale using only a single GPU. It should be noted that the marginally lower loss of BPTT, as mentioned in the original text, was achieved on a small, tractable model, whereas for the GRU model, the online learning method demonstrated superior prediction performance(Please refer to Appendix E).
>
> **Response to Q7:**
>
> This is a valid point; a portion of Drosophila neurons are non-spiking. However, in our approach, we are essentially converting calcium signals into a generalized "firing rate" because neuronal activity, regardless of whether its internal mechanism is graded potentials or action potentials, ultimately leads to changes in intracellular calcium concentration. Although the deconvolved signal may not directly equate to precise electrophysiological parameters of non-spiking neurons, its trajectory over time provides us with crucial information: when the functional output of that neuron significantly increases or decreases. This is sufficient and necessary for parsing the dynamics of neural circuits during behavior.
>
> **Response to Q8:**
>
> We have also noted the discrepancy in the number of strong connections between the learned weight distribution and the real connectome, and we believe this may stem from the demands of dynamics fitting: to reproduce the observed complex neural dynamics within limited model complexity, the model may need to rely on a few strong connections as primary "drivers" of the dynamics. In follow-up work, we will analyze the functional necessity of these strong connections in depth and examine their specific contributions to network dynamics through sensitivity analyses.

---

> > ### Author Response · Authors · 2025-12-02
> >
> > **Response to Q9:**
> >
> > Currently, we can complete training using D-RTRL on a single A40 GPU (with approximately 50GB of memory). To enable a fair comparison, we will implement multi-GPU BPTT training in future work. This would serve as a more meaningful benchmark.
> >
> > **Response to Q10:** We would like to clarify a potential misunderstanding: the x-axis in Fig. 2B represents the **length of the data sequences**, not the training time.
> >
> > **Response to Q11:**
> >
> > Our connectome-constrained model does not show signs of overfitting; on the contrary, the artificial neural network GRU exhibited tendencies of overfitting. Please refer to our response in the "Model performance and baseline evaluation" section.
> >
> > **Response to Q12:**
> >
> > The trained time constants are biologically plausible (Appendix H, Fig. S5). Moreover, most neurons underwent only modest adjustments, with few showing large shifts. This suggests that learning selectively tunes specific subpopulations, potentially enabling the model to capture the multi‑timescale dynamics required for realistic activity patterns.
> >
> > **Response to Q13:**
> >
> > We did attempt to initialize the model weights using the synaptic counts from the connectome. However, this initialization strategy resulted in poor training performance, with the model struggling to converge. This is not surprising, as there is a complex nonlinear mapping between anatomically measured synaptic numbers from EM and the functional weights required to support specific dynamical states. Furthermore, the raw numerical range of synaptic counts may not match the weight scale required by the model, leading to an unstable initial network state. Deep learning practice shows that the initialization scheme is critical for successful optimization, and variance-scaled random initialization (Appendix B) provided a better starting point for optimization in our model.
> >
> > We chose to use connectome information as topological and polarity constraints, rather than as constraints on weight values. Specifically: (1) Topological constraint: the sparsity pattern strictly follows the connectome; (2) Polarity constraint: The sign of each connection is fixed based on neurotransmitter type; (3) Weight magnitude: Learned as free parameters through data-driven optimization.
> >
> > It is worth emphasizing that although we did not explicitly use connectome weights as constraints or for initialization, the optimized model weights spontaneously exhibit similarity to the connectome weight distribution (Figure 4, Appendix I). This result indicates that the combination of functional constraints and structural constraints is sufficient to drive the model to learn a weight distribution pattern consistent with biological observations. This emergent correspondence itself holds significant scientific meaning, suggesting that brain structure and function may jointly adhere to some optimization principle.
> >
> > **Response to Q14:**
> >
> > This is a very important question. Indeed, we could build a simpler mesoscale model that operates directly at the level of the 73 neuropils instead of tracking 138,639 individual neurons. We have tested a GRU-based neuropil-level modeling scheme (Please refer to Appendix F). This approach does significantly reduce model complexity (from ~138,000 neurons to 73 neuropil), but it achieves lower performance in reconstructing neuropil-level activity patterns and functional connectivity (even when model complexity is increased).
> >
> > Simultaneously, this mesoscale modeling paradigm has fundamental limitations. First, it loses single-neuron resolution information, cannot resolve heterogeneity within neuropils, erases the dynamical and computational characteristics of individual neurons. Second, this method cannot fully leverage the structural constraints of the detailed connectome. Synapse-level connection topology is averaged out, the polarity information of excitatory and inhibitory synapses is lost, and the sparse and specific connection patterns within the connectome cannot be modeled, making it difficult to investigate the fine mechanisms of structure-function mapping. Furthermore, mesoscale models limit our insight into neural circuit mechanisms.
> >
> > As global brain science enters the era of whole-brain connectomics and single-neuron recording, the value of single-neuron resolution modeling becomes increasingly prominent. Understanding brain computation requires bridging multiscale mechanisms from synapses to neurons, and from circuits to systems. The single-neuron resolution whole-brain modeling framework proposed in this paper precisely provides the necessary computational infrastructure to leverage these new data. Our model does not aim to replace mesoscale models but rather offers finer resolution when needed, providing a tool to explore how connectome structure shapes single-neuron dynamics, making cross-scale mechanistic studies possible.

---

### Official Review · Reviewer_fvHS · 2025-11-01

**Soundness:** 2
**Presentation:** 2
**Contribution:** 2
**Rating:** 2
**Confidence:** 5

**Summary:**

his paper introduces an online learning framework to train a large-scale, connectome-constrained model of the Drosophila brain. Instead of using backpropagation through time (BPTT), which is computational intensive, the method updates parameters in a forward, online way, allowing training of 130K neurons and millions of synapses on a single GPU. The model fits real resting-state calcium imaging data and reproduces functional connectivity and neural dynamics. Interestingly, it also shows emergent biological features such as heavy-tailed synaptic weights and critical-state dynamics.

**Strengths:**

1. **Motivation:** This work is scientifically grounding, it is rooted in realistic structural constraints (FlyWire connectome) and supported by mechanistic biophysical modeling of neurons and neuropil aggregation.
2. **Method:** It introduces an online learning framework that scales connectome-based neural fitting to whole-brain level, resolving BPTT’s scalability bottleneck, allows effective parameter tuning on a single GPU.
3. **Biological relevance:** This model allows to fit data but reproduces critical-state dynamics, linking structural connectivity, optimized weights, and emergent function in a unified mechanistic system.

**Weaknesses:**

1. **Predictive performance:** While the model reproduces broad activity patterns, there remains a visible mismatch between predicted and ground-truth neural traces, as well as differences in functional connectivity between training and test phases (Fig. 3). This indicates that the fitting may not fully capture detailed neural dynamics.

2. **Generalization:** The work focuses primarily on resting-state data, leaving generalization to stimulus-driven or task-based brain dynamics unexplored.

3. **Scientific insights:** Although the model reproduces criticality and synaptic weight distributions, it does not reveal new mechanistic principles or biological findings beyond reproducing known phenomena. And the presentation and writing style read more like a technical system report than a research paper emphasizing scientific discovery or hypothesis testing.

4. **Baselines**: The paper lacks systematic comparison with alternative modeling frameworks (e.g., mechanistic models, recurrent networks) or other biologically plausible learning rules.

**Questions:**

1. What factors cause the mismatch between training and test functional connectivity (Fig. 3)? Is this due to overfitting, noise in calcium imaging data, or limits of the model’s expressiveness?

2. Could the authors discuss how their framework could be adapted or tested under task-driven or sensory-evoked conditions? This would clarify the general applicability of the method.

3. Beyond reproducing criticality, can the model be used to generate new predictions or hypotheses about how structural connectivity shapes dynamics?

4. Include more implementation details for reproducibility.

---

> ### Author Response · Authors · 2025-12-02
>
> We thank you for reviewing our manuscript and providing valuable feedback. We have carefully considered each of your comments and respond to them point by point below. For clarity, we address several related core issues in the section titled "Model performance and baseline evaluation," which will be referenced in the subsequent specific responses.
>
>
> **Comment:**
>
> - **w1: Predictive performance:** While the model reproduces broad activity patterns, there remains a visible mismatch between predicted and ground-truth neural traces, as well as differences in functional connectivity between training and test phases (Fig. 3). This indicates that the fitting may not fully capture detailed neural dynamics.
> - **w4: Baselines**: The paper lacks systematic comparison with alternative modeling frameworks (e.g., mechanistic models, recurrent networks) or other biologically plausible learning rules.
> - **Q1**:What factors cause the mismatch between training and test functional connectivity (Fig. 3)? Is this due to overfitting, noise in calcium imaging data, or limits of the model’s expressiveness?
> - **Q4**:Include more implementation details for reproducibility.
>
> **Response:**
>
> We thank the reviewer for raising this important question. Please refer to our response in the "Model performance and baseline evaluation" section.
>
> **Comment:**
>
> - **W2: Generalization:** The work focuses primarily on resting-state data, leaving generalization to stimulus-driven or task-based brain dynamics unexplored.
> - **Q2**:Could the authors discuss how their framework could be adapted or tested under task-driven or sensory-evoked conditions? This would clarify the general applicability of the method.
>
> **Response:**
>
> Our model and training paradigm are fully compatible with fitting stimulus-driven or task-based neural activity. The only difference from the current resting-state model is that training under stimulus-driven or task-based conditions would require introducing new parameters to model external stimuli. However, the overall training paradigm remains the same.
>
> Furthermore, the model fitted to resting-state activity can also be used to investigate its dynamical performance under stimulus-driven conditions. For example, similar to (Shiu, P.K. et al., 2024), we applied simulated sugar stimulation to the left gustatory receptor neurons in our model and monitored the subsequent activity of the proboscis motor neurons. The model successfully reproduced a key neurobehavioral phenomenon: activation of appetitive gustatory receptor neurons upon encountering sugar leads to excitation of proboscis motor neurons. Crucially, consistent with established experimental findings, the model showed that the contralateral (right) MN9 neuron exhibited stronger activation than the ipsilateral (left) MN9 neuron. (See Appendix K.)
>
> **Comment:**
>
> - **W3: Scientific insights:** Although the model reproduces criticality and synaptic weight distributions, it does not reveal new mechanistic principles or biological findings beyond reproducing known phenomena. And the presentation and writing style read more like a technical system report than a research paper emphasizing scientific discovery or hypothesis testing.
> - **Q1**:Beyond reproducing criticality, can the model be used to generate new predictions or hypotheses about how structural connectivity shapes dynamics?
>
> **Response:**
>
> Thank you for raising this point. The primary contribution of our current paper is providing an online learning method for connectomics to train unknown biophysical parameters, including synaptic weights, cellular time constants, etc. Our paper emphasizes methodology, and therefore the current content does not include extensive scientific discoveries or hypothesis testing.
>
> It is evident that based on such a functional, connectome-constrained, and interpretable model, various interesting predictions or hypotheses about how structural connectivity shapes dynamics can currently be generated. These include, but are not limited to:how the topological features of the connectome determine criticality;structural strong connections and functional cooperative patterns;structural bottlenecks and slow modes;the co-shaping of intra-module/inter-module weights and time constants;the coordination between the connectome and background inputs (encoding weights).
>
> These topics exceed the scope and length that can be covered in this paper. We will continue to advance and seek more scientific insights.

---

### Author Response · Authors · 2025-12-02
**Model Performance and Baseline Evaluation**

## **Model Performance and Baseline Evaluation**

We thank the reviewers for raising questions regarding the model's predictive performance, overfitting risks, model performance comparisons, and related issues. We provide clarification here.

### **1. Model Paradigm**

As shown in Figure 3, our model performs a generative regression task. We must emphasize the inherently high difficulty of this prediction task itself. The model needs to autoregressively predict the subsequent 750 time steps based on a single initial input, making it highly susceptible to error accumulation effects—where minor initial inaccuracies are continuously amplified over time, leading to significant deviation of the output from the ground truth.

Simultaneously, the inconsistent nature of the data itself further exacerbates this challenge. As shown in Figure 3B, the Drosophila neuropil calcium imaging time series exhibits low intrinsic consistency. The functional connectivity correlation matrix for the first 500 time steps is only 0.474 when compared with that of the last 500 time steps. This further increases the difficulty of prediction after model training.

### **2. Performance of the GRU Model**

To further demonstrate the advantages of our model, we compared it against a strong baseline model—a GRU model with 256 hidden units, fully trained until loss convergence.

Interestingly, we found that training a GRU using BPTT to perform this task does not converge well. In contrast, using the online learning method D-RTRL can train this task effectively. (Please refer to Appendix E and Figure S1.)
**(1)Qualitative Analysis**: As shown in the figures, when faced with unseen test data, the GRU model falls into a local loop, repeating only limited patterns and demonstrating poor generalization ability. In contrast, our model captures richer internal dynamics and can spontaneously generate more diverse and realistic neural activity patterns over extended periods. (Please refer to Appendix F and Figure S2.)
**(2)Quantitative Analysis**: The functional connectivity predicted by the GRU model shows very low correlation with the ground truth (0.211), significantly lower than the performance of our model (0.556) (Please refer to Appendix F and Figure S3). Furthermore, avalanche analysis performed on the GRU output indicates that its activity does not satisfy the power-law distribution commonly observed in neural activity (Please refer to Appendix J and Figure S7).

### **3. Performance of the Model under Whole-Brain Connectome Constraints**

Despite the aforementioned multiple challenges, our model achieved a correlation of **0.556** between the predicted functional connectivity on the test set and the ground truth functional connectivity. It is particularly important to emphasize that **this value is even higher than the intrinsic correlation (0.474) between the earlier and later segments of the data itself.** This indicates that the model has learned generalized dynamical features, rather than merely memorizing the training set.

In summary, given the extraordinary difficulty of this task, we believe these results preliminarily demonstrate the effectiveness and generalization capability of our model.

---

### Author Response · Authors · 2025-12-03
**Summary**

Dear AC,

We sincerely thank you and all reviewers for your valuable time in reviewing our manuscript and for providing insightful and constructive comments. Below, we briefly summarize the core contributions of this work, address key concerns, and reflect on its limitations, hoping to offer a more comprehensive perspective for your final decision.

### **I. Core Value and Innovations of This Study**

We believe this study makes substantial contributions to computational neuroscience (as acknowledged by most reviewers):

1. **Pioneering Model and Scale Breakthrough**: This work presents the first neurally constrained and functionally fitted neurodynamics model at both single-neuron and whole-brain scales based on the actual Drosophila connectome.
2. **Methodological Innovation**: We introduce and validate an online learning-based training framework that successfully circumvents the memory explosion issue of traditional BPTT in large-scale brain simulation, enabling whole-brain model optimization on a single GPU and addressing a key scalability bottleneck in the field.
3. **Model Validation**: We thoroughly validate the framework’s effectiveness. The trained model not only fits neural activity data but also self-organizes to exhibit neural critical dynamics. This demonstrates that our framework links structure and function within a unified, connectome-based system, offering a powerful computational tool for understanding multiscale brain organization.

**In summary, this study overcomes computational bottlenecks in training large-scale, biologically realistic whole-brain models and provides a new paradigm for studying structure–function relationships within a unified framework.**

### **II. Clarifying Key Misunderstandings in the Reviews**

We have provided clearer explanations in the appendix and response letter regarding several misunderstandings, which mainly include:

**1. Model Predictive Performance**

First, we emphasize that the prediction task itself is inherently challenging. The “generative regression task” for neural activity prediction is highly susceptible to error accumulation, leading to significant deviation from ground truth. Second, the experimental data show low consistency across time segments, further increasing the difficulty of learning and prediction. Moreover, even heavily trained GRU models tend to fall into limit cycles during testing, showing poor generalization.

However, we find that the connectome-constrained model generates sustained, rich dynamical patterns with far greater stability than the GRU. On test sets, it outperforms the GRU and even exceeds the temporal consistency of the data itself. Additionally, the connectome-constrained model exhibits resting-state neural avalanches with power-law distributions—a feature not learnable by the GRU.

These results strongly demonstrate that our model learns generalized dynamical features and exhibits excellent performance.

**2. Misunderstanding About Overfitting**

Given the large parameter count of whole-brain models, another concern is overfitting. Through systematic analysis, we show that whether starting from vastly different initializations or training with different random seeds from the same initial conditions, the model consistently converges to highly similar and stable neuropil parameter distributions. This confirms that the model is not overfitted but learns stable parameter patterns from the resting-state dataset.

**3. Relationship Between Resting State and Stimulus-Driven State**

Since we use resting-state data to validate our online learning framework, concerns have been raised regarding differences between resting-state and task-driven paradigms.

Our model and training paradigm are fully compatible with stimulus-driven or task-based neural activity fitting. The only difference from the current resting-state model is that stimulus-driven tasks require additional parameters to model external inputs—but the overall training framework remains the same. Moreover, models trained on resting-state data can also be used to investigate stimulus-driven dynamics, as demonstrated by our model’s successful replication of sugar-sensing feeding behavior consistent with experimental findings.

**4. Misunderstanding About Online Learning vs. BPTT Performance**

Some reviewers questioned the superiority of our online learning algorithm (D-RTRL) over BPTT. To clarify, applying BPTT to our target-scale network (~140k neurons) is infeasible, as even small training batches exceed GPU memory limits. In contrast, D-RTRL runs on a single GPU at this scale. The slightly lower loss reported for BPTT in the manuscript was achieved on a smaller, tractable model, whereas on the GRU model, the online learning approach shows better predictive performance.

---

> ### Author Response · Authors · 2025-12-03
>
> **5. Misunderstanding About Emergent Avalanche Dynamics**
>
> Some reviewers suggested that the critical avalanches in our model might be a general network feature. This is incorrect. The critical dynamics in our model are a learned property, not a universal feature. Our experiments show that a fully trained GRU on the same task does not produce avalanche dynamics following a power-law distribution. Moreover, in the early stages of our model’s training, although avalanches follow a power law, the exponent is very high, indicating near-supercritical dynamics.
>
> **6. Misunderstanding About Initialization Strategy**
>
> Some reviewers questioned our initialization strategy and suggested using EM-based synaptic strengths for initialization. We clarify that we attempted this approach, but the model failed to converge to reasonable loss levels. This may be due to mismatches in scale and distribution of raw EM connectome data, inability of EM data to capture functional heterogeneity, or convergence to poor local optima. Therefore, we adopted the widely validated variance-scaled random initialization from deep learning to ensure stable network activity early in training. Crucially, while weights are randomly initialized, the connectome topology and synaptic polarity are strictly constrained. Interestingly, despite random initialization and no explicit optimization toward EM synaptic strengths, the data-driven optimization spontaneously pushed weight distributions toward statistical properties highly consistent with the experimental connectome.
>
> **7. Misunderstanding About Model Scale**
>
> Some reviewers questioned the necessity of single-neuron-resolution whole-brain modeling and suggested a simpler mesoscale model. We tested GRU-based neuropil-level modeling, which reduces complexity but achieves lower performance in reconstructing neuropil activity patterns and functional connectivity.
>
> Importantly, mesoscale modeling has fundamental limitations. It loses single-neuron resolution and internal heterogeneity within neuropils, erasing individual neuronal dynamics and computational properties. It also fails to leverage fine-grained connectome constraints—synapse-level topology is averaged, synaptic polarity is lost, and sparse, specific connection patterns cannot be modeled, hindering investigation of fine-grained structure–function mapping. Mesoscale models are inadequate for research questions focusing on single-neuron mechanisms.
>
> Understanding brain computation requires bridging mechanisms across scales—from synapses to neurons, circuits, and systems. Our single-neuron-resolution whole-brain framework does not replace mesoscale models but provides finer resolution when necessary, enabling exploration of how connectome structure shapes single-neuron dynamics and facilitating cross-scale mechanistic studies.
>
> **8. Other Misunderstandings**
>
> We have also clarified other misunderstandings, such as the model being a connectome-constrained E/I network rather than a purely excitatory network.
>
> **In summary, regarding misunderstandings about model predictive performance, overfitting risks, generalization to stimulus-driven contexts, and the model itself, we have provided detailed clarifications in the appendix and responses, and hope these do not affect the judgment of this important work’s value.**
>
> ### **III. Limitations of This Study**
>
> We acknowledge the limitations pointed out by reviewers, particularly that this work emphasizes methodological contributions and lacks scientific insights.
>
> Indeed, due to scope and length constraints, we could not fully present scientific insights from the connectome-constrained whole-brain model—such as how connectome topology determines criticality, how strong structural connections correspond to functional co-activation patterns, and how intra- and inter-module weights and time constants co-evolve.
>
> These deeper questions extend beyond the current manuscript. We will continue to build on this work to seek further scientific insights and address these questions in future studies.
>
> ### **IV. Conclusion**
>
> In summary, this study pioneers a new paradigm of “using whole-brain connectomes to predict whole-brain neural activity.” Our online learning method specifically addresses scalability challenges in training large-scale biological neural networks. We believe this work will positively impact neuroscience.
>
> We sincerely thank you and the review team for your diligent work and hope this summary aids your final evaluation. We look forward to receiving your recognition of our work.
>
> Best regards,
>
> The Authors

---

### Meta-Review · Area_Chair_fYRU · 2025-12-09

**Summary:**

Dear authors,

The reviewers-- unanimously and unambigioulsy-- recommended not accepting the papers, with an average score of 2.0. After reading the reviews, your responses, and the paper itself, I am following their recommendation.

I do think that the paper is addressing a highly important and interesting problem, and presents a novel, and promising approach to it. Ultimately, however, it falls short both on demonstrating sufficient evidence that  a clear methodological advance has been achieved, or an interesting neuroscience insight. Moreover, there are several concerns and questions that remain unanswered that would require addressing.

I am sorry that I do not have better news.  I do think this is an important line of work, and hope that you will continue it, and I hope that the the reviewers' feedback will help you in improving the manuscript. There is a _ton_ of material here, both on the methodological side and the neuroscience side that could, if properly expanded upon, make for one (if not more...) interesting papers!

**Reviewer Concerns:**

Please see the reviewers' points for some examples of open questions, I am just picking out some highlights here:
- the coupling of each neurons' activity to the 'population state' seems to be a major component of the model, but a major departure from the connectome constraints.  It is unclear if and how that affects the resulting dynamics, and no ablation studies are presented.
- the model seems vastly over-paramterized, and reviewers were not convinced it is not overfitting.
- There were also some questions about whether the comparisons with BPPT were fair. In particular, it seems that they compared to 'full' BPPT, without considering truncated or check-pointed variants.
In general, I think that the reviews, in particular by reviews 2GYa are appropriate, so please take this into account.

**Reviewer Scores:**

I am unwilling to follow this request. The ACs are being asked to perform an unreasonable amount of extra work this year. In particular, I understand the concern about potential collusion. However,  I do think that asking ACs to now read a whole new set of papers, reviews and (extremely long...) rebuttals and to try to condense them into decisions without a chance for discussions is both asking a lot from us, and will, inevitably, result in (on average) poorer decisions for _everyone_.

I do not see any value in spending even more time to try to 'guess' how each reviewer would have changed their mind or not. I am just trying to make appropriate decisions on the paper and focus on the science. The authors can see the reviewer inputs and should take it into account.

---

### Decision · Program_Chairs · 2026-01-26

Reject